# Ranked Entropy Minimization for Continual Test-Time Adaptation

**Jisu Han** [1]    **Jaemin Na** [2]    **Wonjun Hwang** [3]

## Abstract

Test-time adaptation aims to adapt to realistic environments in an online manner by learning during test time. Entropy minimization has emerged as a principal strategy for test-time adaptation due to its efficiency and adaptability. Nevertheless, it remains underexplored in continual test-time adaptation, where stability is more important. We observe that the entropy minimization method often suffers from model collapse, where the model converges to predicting a single class for all images due to a trivial solution. We propose ranked entropy minimization to mitigate the stability problem of the entropy minimization method and extend its applicability to continuous scenarios. Our approach explicitly structures the prediction difficulty through a progressive masking strategy. Specifically, it gradually aligns the model's probability distributions across different levels of prediction difficulty while preserving the rank order of entropy. The proposed method is extensively evaluated across various benchmarks, demonstrating its effectiveness through empirical results. Our code is available at https://github.com/pilsHan/rem

## 1. Introduction

The real world is non-i.i.d., which demands real-time adaptation of AI applications. Deep learning models have achieved remarkable progress in recent years; however, performance degradation caused by distribution shifts between different domains limits the generalization capabilities (Shimodaira, 2000). Test-time adaptation (TTA) (Wang et al., 2021) has emerged as a practical approach to address non-stationary environmental changes by enabling models trained on source domain to adapt in an online manner to unlabeled target data during the test-time.

[1]Ajou University [2]Korea Telecom [3]Korea University. Correspondence to: Wonjun Hwang <wjhwang@korea.ac.kr>.

*Proceedings of the 42nd International Conference on Machine Learning*, Vancouver, Canada. PMLR 267, 2025. Copyright 2025 by the author(s).

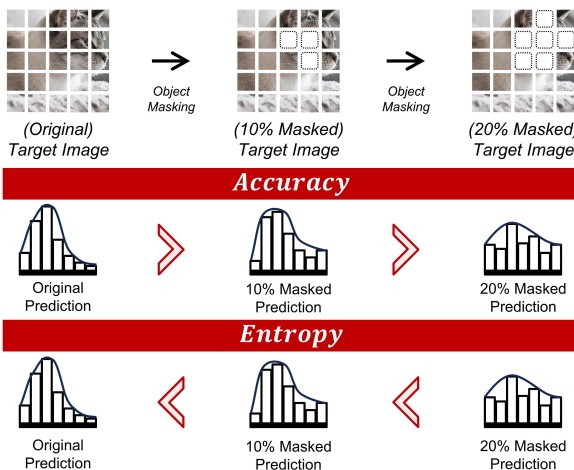

*Figure 1.* **Our Intuition.** We explicitly raise the prediction difficulty of the input images through the masking strategy. Based on the intuition that increased difficulty decreases prediction accuracy and increases entropy, we attempt to maintain a rank ordering of entropy while improving consistency from original to masked predictions. Our approach addresses the problem of model collapse in entropy minimization methods in a simple yet efficient way.

Continual test-time adaptation (CTTA) (Wang et al., 2022) addresses the issue of error accumulation in long-sequence domains. It mitigates the forgetting problem under continuous environmental changes and sequentially adapts to a stream of data, facilitating the practical deployment of TTA. Recent studies on CTTA can be broadly categorized into two major approaches: entropy minimization (EM) (Wang et al., 2021; Niu et al., 2022; Zhang et al., 2025) and consistency regularization (CR) (Wang et al., 2022; Liu et al., 2024b;a). The EM approach minimizes the entropy of predictions and offers computational efficiency but suffers from instability due to the risk of trivial solutions, where predictions collapse into a single class. In contrast, the CR approach employs a teacher-student framework (Tarvainen & Valpola, 2017), updating the model conservatively to ensure stability but incurs high computational costs. As a result, there is a trade-off between efficiency and stability in these approaches.

**Motivation.** Our approach stems from the intuitive observation that a model's predictions for an explicitly information-degraded image are inaccurate and have high entropy compared to its predictions for a complete image (as illustrated

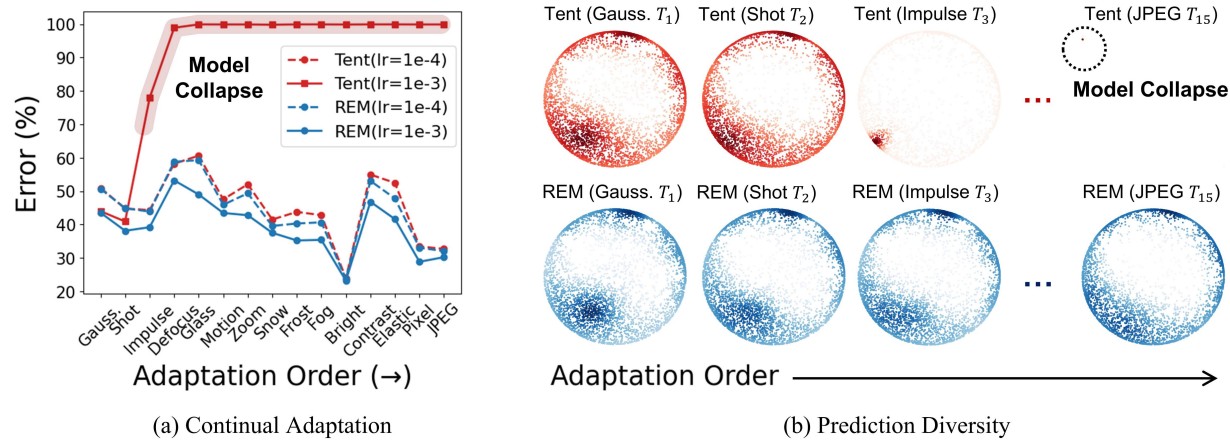

(a) Continual Adaptation          (b) Prediction Diversity

*Figure 2.* **Observation on model collapse in the entropy minimization approach.** (a) Under the CTTA scenario, the EM approach (**Tent**) undergoes significant performance degradation at a critical point (adaptation order $T_3$, Impulse noise). (b) This phenomenon occurs because the model learns constant representations that do not depend on input images, leading to a collapse in prediction diversity. This is evidenced by class probabilities converging to a single point when visualized in a polar coordinate system. Our proposed method (**REM**) mitigates model collapse and maintains prediction diversity.

in Figure 1). Motivated by Zeno's Achilles and the tortoise paradox (Huggett, 2002), this paper aims to prevent abrupt entropy reduction by gradually advancing the original predictions while reducing the gaps in explicit rank relationships. Here, an original prediction is represented by the tortoise, maintaining its leading position over masked prediction in the ranking relationship and learning through rank regularization. In contrast, masked prediction is analogous to Achilles, aiming to catch up with original prediction by learning through consistency regularization.

**Observation.** Figure 2 shows the phenomenon of model collapse in EM. Model collapse means that the EM approach converges to a prediction for a constant class by a trivial solution, resulting in the model to produce a nonsensical prediction. This issue corresponds to mode collapse in GAN (Goodfellow et al., 2014) and complete collapse (Hua et al., 2021) in self-supervised learning (SSL) within the CTTA. Model collapse occurs because the EM objective function is minimized even when the model consistently predicts a single class, regardless of the input. Since EM lacks stability to address the forgetting problem, it maintains performance by using a low learning rate. However, this leads to limitations in both adaptability and robustness.

Based on this observation, we propose a **Ranked Entropy Minimization (REM)**. Specifically, we exploit the self-attention structure of ViT (Dosovitskiy et al., 2021) to mask patches with a high likelihood of containing objects (Bolya et al., 2023; Son et al., 2024). The principal idea is to explicitly enhance the prediction complexity of a sample by masking objects that domain invariant features. Building a mask chain that sequentially obscures more patches based on the masking ratio transforms unpredictable prediction

tendencies into a ranked predictable one. For taking advantage of the ranked structure, we provide two interrelated methods. (1) First, we apply a consistency loss by ensuring that predictions with a higher masking ratio are similar to those with a lower masking ratio, thereby not only indirectly reducing prediction entropy but also enabling the model to learn contextual information from masked regions. (2) Second, we introduce a ranking loss that ensures the entropy of predictions with a lower masking ratio remains lower than that of predictions with a higher masking ratio. This approach not only models uncertainty but also reduces entropy by incorporating object-specific information, preventing predictions from being biased toward background information. The main benefit of our method lies in achieving the joint goals of stability and adaptability while maintaining the efficiency of EM approaches with a single model and without requiring additional models.

## 2. Related Work

### 2.1. Test-Time Adaptation

The concept of optimizing during test time to adapt to target domains was proposed in Test-Time Training (TTT) (Sun et al., 2020). However, TTT relies on the use of source data and training loss, which may not generalize well to practical applications. To address these limitations, Tent (Wang et al., 2021) proposed the concept of Fully TTA, which enables online adaptation to unlabeled target data without requiring access to source data. In this context, TTA focuses on efficiency, typically by updating the normalization layers (Gong et al., 2022) or adjusting predictions without training the model parameters to improve computational efficiency. (Iwasawa & Matsuo, 2021; Boudiaf et al., 2022).

Entropy minimization approaches (Wang et al., 2021; Niu et al., 2022; 2023; Lee et al., 2024; Zhang et al., 2025) are evolving as a primary solution for TTA. Among those, EATA (Niu et al., 2022) introduces sample filtering for uncertain predictions and regularization method to enhance stability. Furthermore, SAR (Niu et al., 2023) observes the phenomenon of model collapse by trivial solutions, and proposes adaptation to flat minima and filtering of large gradient samples. Building on these approaches, DeYO (Lee et al., 2024) presents a criteria for additional sample selection through image rearrangement, and COME (Zhang et al., 2025) proposes a conservative entropy minimization method to address the overconfidence problem.

### 2.2. Continual Test-Time Adaptation

Advancing from single-domain adaptation, CTTA (Wang et al., 2022; Brahma & Rai, 2023; Döbler et al., 2023; Liu et al., 2024b;a) emphasizes the requirement for adapting to continuous domain shifts, thereby extending the practicality of TTA. This paradigm promotes rethinking the conventional protocol by encouraging a deeper focus on stability. CoTTA (Wang et al., 2022) addresses catastrophic forgetting by introducing a consistency loss between the base model and weight-averaged model, while employing stochastic restoration of parameters based on the source model. PETAL (Brahma & Rai, 2023) introduces a probabilistic framework for CTTA and a parameter restoration method leveraging the Fisher Information Matrix. ViDA (Liu et al., 2024b) presents a trade-off between stability and plasticity by designing an adapter that explicitly separates domain-invariant features and domain-specific features. Continual-MAE (Liu et al., 2024a) measures pixel uncertainty through Monte Carlo (MC) dropout to distinguish object presence and enhances the representation of domain-invariant properties using a masked autoencoder.

**Beyond not Forgetting.** The recent state-of-the-art CTTA methods achieve stability and mitigate performance degradation by adopting teacher-student frameworks or parameter restoration from the source model. However, these approaches often neglect efficiency constraints, leading to excessive computational costs and memory overhead. This inefficiency contradicts the goal of TTA, which is to enable real-time adaptation under resource constraints. To address this issue and realign the framework with its intended purpose, protocols that account for computational time constraints (Alfarra et al., 2024) and label delays (Csaba et al., 2024) have been proposed. We argue that efficiency in CTTA requires renewed attention and propose leveraging insights from pioneering approaches in TTA to improve both stability and computational efficiency. In this study, we integrate the strengths of EM and CR approaches to balance efficiency and stability. Detailed efficiency analysis is provided in Section 4.4.

## 3. Ranked Entropy Minimization

In this section, we introduce the CTTA setup and the trivial solution from the EM, then explain the proposed REM. Our method consists of a mask chaining strategy and two loss functions for CR and EM. Masked consistency loss ensures consistency from predictions with a lower masking ratio to those with a higher masking ratio, while entropy ranking loss enforces a ranking constraint from predictions with a higher masking ratio to those with a lower masking ratio.

### 3.1. Preliminaries

In CTTA setup, a target model $f_t$ is trained in an online manner, starting from a source model $f_s$. The model adapts to sequential target domain data $x_t^\tau$, where each domain is represented in order by $\tau \in \{1, 2, \dots\}$. The evaluation protocol involves calculating the cumulative error based on the model's predictions $\hat{p}_t^\tau = f_t(x_t^\tau)$, while the model adapts to the target domains during test time using $x_t^\tau$. The main difference between TTA and CTTA lies in whether the model is reset to the source model when a domain changes. In CTTA, the model is not reset, which makes the issue of catastrophic forgetting more significant.

The EM method applies gradient descent using entropy $\mathcal{S}(\hat{p}_t) = -\sum_{c=1}^{C} \hat{p}_{t,c} \log \hat{p}_{t,c}$ over the total number of classes $C$ as the objective function. Consequently, the gradient of $\mathcal{S}(\hat{p}_t)$ with respect to the parameter $\theta$ is as follows:

$$\frac{\partial \mathcal{S}(\hat{p}_t)}{\partial \theta_t} = -\sum_{i=1}^{C} \left(\log \hat{p}_{t,c} + 1\right) \hat{p}_{t,c}(1 - \hat{p}_{t,c}) \frac{\partial z_{t,c}}{\partial \theta_t}, \quad (1)$$

where $z_{t,c}$ represents the logit for class $c$ before applying the softmax function, i.e., $\hat{p}_{t,c} = \frac{\exp(z_{t,c})}{\sum_{i=1}^{C} \exp(z_{t,i})}$. The trivial solution for $\frac{\partial \mathcal{S}(\hat{p}_t)}{\partial \theta_t} = 0$ arises under the following cases: **(Case 1)** Uniformly distributed probability ($\hat{p}_{t,c} = \frac{1}{C}$), and **(Case 2)** Perfectly confident predictions ($\hat{p}_{t,c} \in \{0, 1\}$). In this case, since the initial source model's prediction distribution is not uniform, the likelihood of achieving a trivial solution due to a uniform distribution under entropy minimization is low. However, it is possible for a singular class to result in perfectly confident predictions, which is experimentally observed in Figure 2.

### 3.2. Explicit Mask Chaining

Conventional CTTA methods point out that augmentation policies may not be valid for dramatic domain shifts and decide whether to apply augmentation through prediction confidence (Wang et al., 2022). However, these methods fail to model the impact of augmentation policies on model predictions, resulting in inefficiencies caused by the application of diverse augmentations. In order to model the predictions, we propose Explicit Mask Chaining, which incrementally

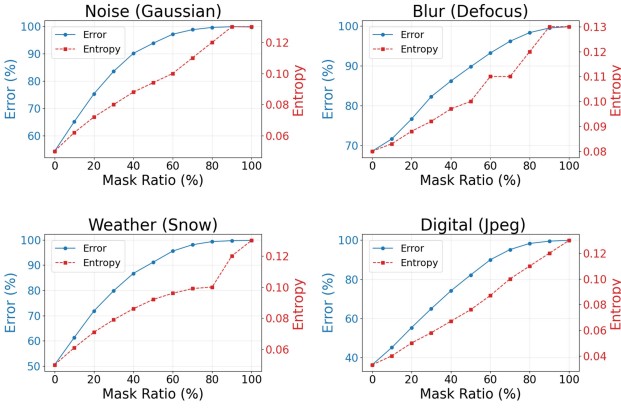

*Figure 3.* **Empirical study according to masking ratio.** We report the changes in error and entropy as the masking ratio increases. Both entropy and error exhibit a monotone increasing trend with respect to the masking ratio, and we observe that linearity becomes more pronounced, especially in regions with lower masking ratios.

masks content containing domain-invariant information for augmentation. To this end, we focus on the self-attention structure of ViT for efficient object masking. Exploiting the self-attention structure allows us to cluster content by similarity between intermediate tokens (Bolya et al., 2023), as well as compute an attention score for an attention head (Son et al., 2024). Following pioneering research, we define an attention score $A$ as follows:

$$A = \sum_{h=1}^{H} \text{Softmax}\left(\frac{Q_{h,cls}K_{h,img}^{\top}}{\sqrt{d}}\right), \qquad (2)$$

where $H$ is the number of heads in the multi-head attention, $Q_{h,cls}$ and $K_{h,img}$ are query for class token and key for image token of the $h$-th attention head, respectively. $d$ denotes the dimension of each attention head.

The masked image $\{x_{m_1}, x_{m_2}, \cdots, x_{m_N}\}$, defined for the top-$m$ proportion of a set $A$ sorted in descending order, has mask ratios that satisfy the condition $0 \leq m_1 \leq m_2 \leq \cdots \leq m_N \leq 1$. Figure 3 shows the error and entropy across a range of masking ratios for noise, blur, weather, and digital corruptions on ImageNetC (Hendrycks & Dieterich, 2018). The results empirically confirm that explicit masking of objects results in lower accuracy and higher entropy as the masking ratio increases. In conclusion, we establish an explicit ranking relationship between entropy and accuracy by employing an incremental masking strategy for consecutive mask ratio.

### 3.3. Masked Consistency Loss

Given an explicitly ranked prediction, we aim to make a high masking rate prediction with relatively low accuracy similar to a low mask rate prediction with relatively high

accuracy. We define masked consistency loss (MCL) as follows:

$$\mathcal{L}_{MCL} = \sum_{i<j}^{M_N} \mathcal{H}(f_t(x_j), \mathbf{sg}(f_t(x_i))), \qquad (3)$$

where $\mathcal{H}(p, q)$ denotes the cross-entropy between two probability distributions $p$ and $q$, $M_N = \{0, m_1, m_2, \ldots, m_N\}$ is a set of mask ratios, where $N$ is the number of mask chains, and $\mathbf{sg}$ denotes the stop-gradient operation.

**Comparison with EM.** Unlike EM, which uses entropy as a loss function, MCL indirectly reduces prediction entropy by using the cross-entropy between masked predictions and either less-masked or unmasked predictions. This is designed to mitigate abrupt entropy changes and overconfident predictions, leading to the alleviation of model collapse.

**Comparison with CR.** To generate stable predictions distinct from the target model, the CR approach requires additional teacher models and numerous forward passes for uncertainty estimation. In contrast, our method eliminates the need for uncertainty prediction by leveraging an explicit ranking structure and improves efficiency by generating diverse predictions via mask chains within a single model.

### 3.4. Entropy Ranking Loss

Through the reduction of differences in ranked prediction distributions, MCL is designed to indirectly minimize prediction entropy. However, this may result in slower adaptation due to small differences in prediction distributions or lead to biased predictions toward the background when learning from images with occluded objects. To complement MCL and address these issues, we propose the entropy ranking loss (ERL) as follows:

$$\mathcal{L}_{ERL} = \sum_{i<j}^{M_N} \max\left(0, S(f_t(x_i)) - \mathbf{sg}(S(f_t(x_j))) + \mathsf{m}\right), \qquad (4)$$

where $\mathsf{m}$ is the margin. The purpose of ERL is to maintain the principle that the entropy of predictions with a low masking ratio for objects should be lower than that of predictions with a high masking ratio. By maintaining an explicit ranked order, we prevent overconfidence in high masking ratio predictions, which could lead to bias toward the background. This follows from prior findings that applying ranking losses in neural calibration effectively mitigates overconfidence (Moon et al., 2020; Noh et al., 2023). Additionally, ERL directly reduces the entropy of samples that violate the ranked order, thereby promoting faster adaptation and maintaining a structured entropy hierarchy across different masking ratios.

*Table 1.* Classification error rate (%) for ImageNet-to-ImageNetC under CTTA scenario. Mean (%) denotes the average error rate across 15 target domains. Gain (%) represents the relative performance improvement compared to the source model.

| Time $t$ → | | | | | | | | | | | | | | | | | |
| Method | Gaussian | shot | impulse | defocus | glass | motion | zoom | snow | frost | fog | brightness | contrast | elastic_trans | pixelate | jpeg | Mean↓ | Gain |
| --- | --- | --- | --- | --- | --- | --- | --- | --- | --- | --- | --- | --- | --- | --- | --- | --- | --- |
| Source (Dosovitskiy et al., 2021) | 53.0 | 51.8 | 52.1 | 68.5 | 78.8 | 58.5 | 63.3 | 49.9 | 54.2 | 57.7 | 26.4 | 91.4 | 57.5 | 38.0 | 36.2 | 55.8 | 0.0 |
| Pseudo-label (Lee, 2013) | 45.2 | 40.4 | 41.6 | 51.3 | 53.9 | 45.6 | 47.7 | 40.4 | 45.7 | 93.8 | 98.5 | 99.9 | 99.9 | 98.9 | 99.6 | 61.2 | -5.4 |
| Tent (Wang et al., 2021) | 52.2 | 48.9 | 49.2 | 65.8 | 73.0 | 54.5 | 58.4 | 44.0 | 47.7 | 50.3 | 23.9 | 72.8 | 55.7 | 34.4 | 33.9 | 51.0 | +4.8 |
| CoTTA (Wang et al., 2022) | 52.9 | 51.6 | 51.4 | 68.3 | 78.1 | 57.1 | 62.0 | 48.2 | 52.7 | 55.3 | 25.9 | 90.0 | 56.4 | 36.4 | 35.2 | 54.8 | +1.0 |
| VDP (Gan et al., 2023) | 52.7 | 51.6 | 50.1 | 58.1 | 70.2 | 56.1 | 58.1 | 42.1 | 46.1 | 45.8 | 23.6 | 70.4 | 54.9 | 34.5 | 36.1 | 50.0 | +5.8 |
| SAR (Niu et al., 2023) | 49.3 | 43.8 | 44.9 | 58.2 | 60.9 | 46.1 | 51.8 | 41.3 | 44.1 | 41.8 | 23.8 | 57.2 | 49.9 | 32.9 | 32.7 | 45.2 | +10.6 |
| PETAL (Brahma & Rai, 2023) | 52.1 | 48.2 | 47.5 | 66.8 | 74.0 | 56.7 | 59.7 | 46.8 | 47.2 | 52.7 | 26.4 | 91.3 | 50.7 | 32.3 | 32.0 | 52.3 | +3.5 |
| ViDA (Liu et al., 2024b) | 47.7 | 42.5 | 42.9 | 52.2 | 56.9 | 45.5 | 48.9 | 38.9 | 42.7 | 40.7 | 24.3 | 52.8 | 49.1 | 33.5 | 33.1 | 43.4 | +12.4 |
| Continual-MAE (Liu et al., 2024a) | 46.3 | 41.9 | 42.5 | **51.4** | 54.9 | **43.3** | **40.7** | **34.2** | 35.8 | 64.3 | 23.4 | 60.3 | **37.5** | 29.2 | 31.4 | 42.5 | +13.3 |
| REM (Ours) | **43.5** | **38.1** | **39.2** | 53.2 | **49.0** | 43.5 | 42.8 | 37.5 | **35.2** | **35.4** | **23.2** | **46.8** | 41.6 | **28.9** | **30.2** | **39.2** | **+16.6** |
| Supervised | 42.5 | 36.9 | 37.1 | 46.4 | 44.0 | 37.4 | 38.3 | 34.2 | 33.1 | 32.5 | 21.5 | 43.3 | 34.4 | 26.1 | 27.5 | 35.7 | +20.1 |

*Table 2.* Classification error rate (%) for CIFAR10-to-CIFAR10C under CTTA scenario. Mean (%) denotes the average error rate across 15 target domains. Gain (%) represents the relative performance improvement compared to the source model.

| Time $t$ → | | | | | | | | | | | | | | | | | |
| Method | Gaussian | shot | impulse | defocus | glass | motion | zoom | snow | frost | fog | brightness | contrast | elastic_trans | pixelate | jpeg | Mean↓ | Gain |
| --- | --- | --- | --- | --- | --- | --- | --- | --- | --- | --- | --- | --- | --- | --- | --- | --- | --- |
| Source (Dosovitskiy et al., 2021) | 60.1 | 53.2 | 38.3 | 19.9 | 35.5 | 22.6 | 18.6 | 12.1 | 12.7 | 22.8 | 5.3 | 49.7 | 23.6 | 24.7 | 23.1 | 28.2 | 0.0 |
| Pseudo-label (Lee, 2013) | 59.8 | 52.5 | 37.2 | 19.8 | 35.2 | 21.8 | 17.6 | 11.6 | 12.3 | 20.7 | 5.0 | 41.7 | 21.5 | 25.2 | 22.1 | 26.9 | +1.3 |
| Tent (Wang et al., 2021) | 57.7 | 56.3 | 29.4 | 16.2 | 35.3 | 16.2 | 12.4 | 11.0 | 11.6 | 14.9 | 4.7 | 22.5 | 15.9 | 29.1 | 19.5 | 23.5 | +4.7 |
| CoTTA (Wang et al., 2022) | 58.7 | 51.3 | 33.0 | 20.1 | 34.8 | 20 | 15.2 | 11.1 | 11.3 | 18.5 | 4.0 | 34.7 | 18.8 | 19.0 | 17.9 | 24.6 | +3.6 |
| VDP (Gan et al., 2023) | 57.5 | 49.5 | 31.7 | 21.3 | 35.1 | 19.6 | 15.1 | 10.8 | 10.3 | 18.1 | 4.0 | 27.5 | 18.4 | 22.5 | 19.9 | 24.1 | +4.1 |
| SAR (Niu et al., 2023) | 54.1 | 47.6 | 38.0 | 19.9 | 34.8 | 22.6 | 18.6 | 12.1 | 12.7 | 22.8 | 5.3 | 39.9 | 23.6 | 24.7 | 23.1 | 26.6 | +1.6 |
| PETAL (Brahma & Rai, 2023) | 59.9 | 52.3 | 36.1 | 20.1 | 34.7 | 19.4 | 14.8 | 11.5 | 11.2 | 17.8 | 4.4 | 29.6 | 17.6 | 19.2 | 17.3 | 24.4 | +3.8 |
| ViDA (Liu et al., 2024b) | 52.9 | 47.9 | 19.4 | 11.4 | 31.3 | 13.3 | 7.6 | 7.6 | 9.9 | 12.5 | 3.8 | 26.3 | 14.4 | 33.9 | 18.2 | 20.7 | +7.5 |
| Continual-MAE (Liu et al., 2024a) | 30.6 | 18.9 | 11.5 | 10.4 | 22.5 | 13.9 | 9.8 | **6.6** | 6.5 | 8.8 | 4.0 | 8.5 | 12.7 | 9.2 | 14.4 | 12.6 | +15.6 |
| REM (Ours) | **17.3** | **12.5** | **10.3** | **8.4** | **17.7** | **8.4** | **5.5** | **6.6** | **5.6** | **7.2** | **3.7** | **6.4** | **11.0** | **7.3** | **13.0** | **9.4** | **+18.8** |
| Supervised | 14.6 | 9.0 | 6.9 | 6.1 | 11.2 | 6.0 | 3.7 | 4.4 | 3.4 | 4.9 | 2.1 | 3.7 | 7.5 | 4.3 | 8.5 | 6.4 | +21.8 |

### 3.5. Total Loss Function

The total loss function $\mathcal{L}_{REM}$ is expressed as a linear combination of $\mathcal{L}_{MCL}$ in Eq. 3 and $\mathcal{L}_{ERL}$ in Eq. 4, as follows:

$$\mathcal{L}_{REM} = \mathcal{L}_{MCL} + \lambda \cdot \mathcal{L}_{ERL}, \qquad (5)$$

where $\lambda$ is a hyperparameter. We propose Ranked Entropy Minimization, which integrates the advantages of consistency regularization and entropy minimization within a ranked structure based on masking ratios.

## 4. Experiments

In this section, we extensively explore the effectiveness of our REM on CTTA protocol (Wang et al., 2022). The analysis includes comparisons with state-of-the-art baselines, verification of its intended functionality through visualizations, and an understanding of its working mechanisms through ablation studies. Additionally, experiments on Online TTA and Vision-Language Model are provided in Appendix C, D.

### 4.1. Experimental Setup

**Benchmarks.** We construct experiments on ImageNet-to-ImageNetC, CIFAR10-to-CIFAR10C, and CIFAR100-to-CIFAR100C. The source domains are ImageNet (Deng et al., 2009) and CIFAR (Krizhevsky et al., 2009), while the corresponding robustness benchmarks (Hendrycks & Dietterich, 2018), ImageNetC, CIFAR10C, and CIFAR100C, are used as the target domains. The suffix C in these datasets indicates corruption, which includes 15 types of corruptions, each with 5 levels of severity. Following (Wang et al., 2022; Liu et al., 2024b;a), we adopt target domains with level 5 severity across all 15 corruption types for sequential domains. We evaluate the classification error rate for each target domain after adaptation and prediction on target domain data streams in an online manner.

**Comparison Methods.** We compare various types of state-of-the-art CTTA approaches using the ViT-B/16 (Dosovitskiy et al., 2021) pre-trained on the source domain. These include single model-based methods such as Pseudo-label (Lee, 2013), Tent (Wang et al., 2021), VDP (Gan et al., 2023), and SAR (Niu et al., 2023), as well as teacher-student frameworks, including CoTTA (Wang et al., 2022), PETAL (Brahma & Rai, 2023), ViDA (Liu et al., 2024b), and Continual-MAE (Liu et al., 2024a). Additionally, the supervised results, obtained by training with target labels using cross-entropy loss, are presented as an upper bound since target labels are unavailable in TTA.

*Table 3.* Classification error rate (%) for CIFAR100-to-CIFAR100C under CTTA scenario. Mean (%) denotes the average error rate across 15 target domains. Gain (%) represents the relative performance improvement compared to the source model.

| Time | $t$ → | | | | | | | | | | | | | | | | |
|---|---|---|---|---|---|---|---|---|---|---|---|---|---|---|---|---|---|
| Method | Gaussian | shot | impulse | defocus | glass | motion | zoom | snow | frost | fog | brightness | contrast | elastic_trans | pixelate | jpeg | Mean↓ | Gain |
| Source (Dosovitskiy et al., 2021) | 55.0 | 51.5 | 26.9 | 24.0 | 60.5 | 29.0 | 21.4 | 21.1 | 25.0 | 35.2 | 11.8 | 34.8 | 43.2 | 56.0 | 35.9 | 35.4 | 0.0 |
| Pseudo-label (Lee, 2013) | 53.8 | 48.9 | 25.4 | 23.0 | 58.7 | 27.3 | 19.6 | 20.6 | 23.4 | 31.3 | 11.8 | 28.4 | 39.6 | 52.3 | 33.9 | 33.2 | +2.2 |
| Tent (Wang et al., 2021) | 53.0 | 47.0 | 24.6 | 22.3 | 58.5 | 26.5 | 19.0 | 21.0 | 23.0 | 30.1 | 11.8 | 25.2 | 39.0 | 47.1 | 33.3 | 32.1 | +3.3 |
| CoTTA (Wang et al., 2022) | 55.0 | 51.3 | 25.8 | 24.1 | 59.2 | 28.9 | 21.4 | 21.0 | 24.7 | 34.9 | 11.7 | 31.7 | 40.4 | 55.7 | 35.6 | 34.8 | +0.6 |
| VDP (Gan et al., 2023) | 54.8 | 51.2 | 25.6 | 24.2 | 59.1 | 28.8 | 21.2 | 20.5 | 23.3 | 33.8 | **7.5** | **11.7** | 32.0 | 51.7 | 35.2 | 32.0 | +3.4 |
| SAR (Niu et al., 2023) | 39.4 | 31.0 | 19.8 | 20.9 | 43.9 | 22.6 | 19.1 | 20.3 | 20.2 | 24.3 | 11.8 | 22.3 | 35.2 | 32.1 | 30.1 | 26.2 | +9.2 |
| PETAL (Brahma & Rai, 2023) | 49.2 | 38.7 | 24.1 | 26.3 | 38.2 | 25.4 | 19.4 | 21.0 | 19.3 | 26.6 | 15.4 | 31.8 | 28.3 | 26.6 | 29.5 | 28.0 | +7.4 |
| ViDA (Liu et al., 2024b) | 50.1 | 40.7 | 22.0 | 21.2 | 45.2 | 21.6 | **16.5** | **17.9** | **16.6** | 25.6 | 11.5 | 29.0 | **29.6** | 34.7 | **27.1** | 27.3 | +8.1 |
| Continual-MAE (Liu et al., 2024a) | 48.6 | 30.7 | 18.5 | 21.3 | 38.4 | 22.2 | 17.5 | 19.3 | 18.0 | 24.8 | 13.1 | 27.8 | 31.4 | 35.5 | 29.5 | 26.4 | +9.0 |
| REM (Ours) | **29.2** | **25.5** | **17.0** | **19.1** | **35.2** | **21.2** | 18.3 | 19.5 | 18.7 | **22.8** | 15.5 | 17.6 | 31.6 | **26.2** | 33.0 | **23.4** | **+12.0** |
| Supervised | 26.2 | 20.6 | 13.9 | 15.9 | 24.6 | 15.6 | 11.8 | 13.1 | 12.1 | 13.6 | 8.5 | 9.7 | 20.2 | 13.5 | 21.5 | 16.1 | +19.3 |

*Table 4.* Forward transfer analysis on ImageNetC. Results (%) represent the error rates for unseen and seen domains, harmonic mean.

| Method | Directly test on unseen domains | | | | | Unseen | Seen | Harmonic |
|---|---|---|---|---|---|---|---|---|
| | bri. | contrast | elastic | pixelate | jpeg | Mean↓ | Mean↓ | Mean↓ |
| Source | 26.4 | 91.4 | 57.5 | 38.0 | 36.2 | 49.9 | 58.8 | 54.0 |
| Tent | 25.8 | 91.9 | 57.0 | 37.2 | 35.7 | 49.5 | 54.4 | 51.8 |
| CoTTA | 25.3 | 88.1 | 55.7 | 36.4 | 34.6 | 48.0 | 57.8 | 52.4 |
| ViDA | 24.6 | 68.2 | 49.8 | **34.7** | 34.1 | 42.3 | 45.9 | 44.0 |
| REM (lr=1e-4) | **23.9** | **66.3** | **47.6** | 35.9 | **33.1** | **41.4** | 45.4 | **43.1** |
| REM (lr=1e-3) | 24.8 | 66.9 | 53.5 | 40.0 | 39.4 | 44.9 | **42.1** | 43.5 |
| Supervised | 22.5 | 71.1 | 55.3 | 38.2 | 36.6 | 44.7 | 38.2 | 41.2 |

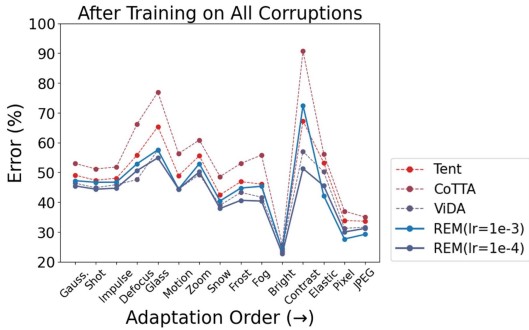

*Figure 4.* Backward transfer analysis on ImageNetC. We compare the performance of CTTA approaches on previous domains.

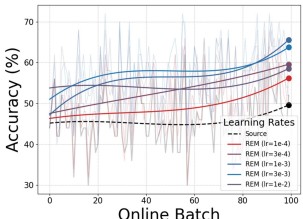
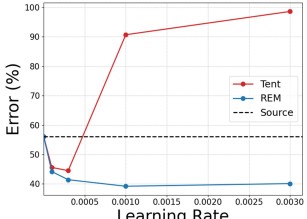

*Figure 5.* Adaptability analysis on ImageNetC under Gaussian noise corruption.

*Figure 6.* Robustness analysis with respect to learning rate on ImageNetC.

**Implementation.** We implemented the experiments on CIFAR10C and CIFAR100C using the open-source Continual-MAE code and the provided source model weights. For ImageNetC, the experiments were conducted using the open-source ViDA code with ImageNet pre-trained weights from timm (Wightman, 2019). Details for reproducibility and training regimes are provided in Appendix B.

## 4.2. Quantitative Results

**ImageNet-to-ImageNetC.** Table 1 presents the CTTA experimental results for a source model pre-trained on ImageNet, using each corruption in ImageNetC as the target domain. The model sequentially adapts to the target domains over time, and we compare the average error for each domain. Our method improves average performance by 16.6% over the source model and surpasses the previous state-of-the-art Continual-MAE by 3.3%. Notably, the performance gap to the supervised learning upper bound is only 3.5%, demonstrating the effectiveness of our approach.

**CIFAR10-to-CIFAR10C and CIFAR100-to-CIFAR100C.** Table 2 and 3 summarize the experimental results for models trained on CIFAR10 and CIFAR100 as source domains, with CIFAR10C and CIFAR100C serving as the respective target domains. The results reveal consistent performance improvements on CIFAR10C and CIFAR100C, which are widely recognized benchmarks for CTTA alongside Ima-

geNetC. These findings highlight the robustness and adaptability of our method across diverse datasets.

**Forward and Backward Transfer Analysis.** We analyze the performance on both future and past domains during the CTTA process to investigate the potential temporal effects of domain adaptation. Table 4 presents the performance on the 5 unseen domains after training on 10 domains. The investigation of the impact of test-time adaptation at the current time on future performance reveals a trade-off between adaptability and generalization. Rapid adaptation driven by

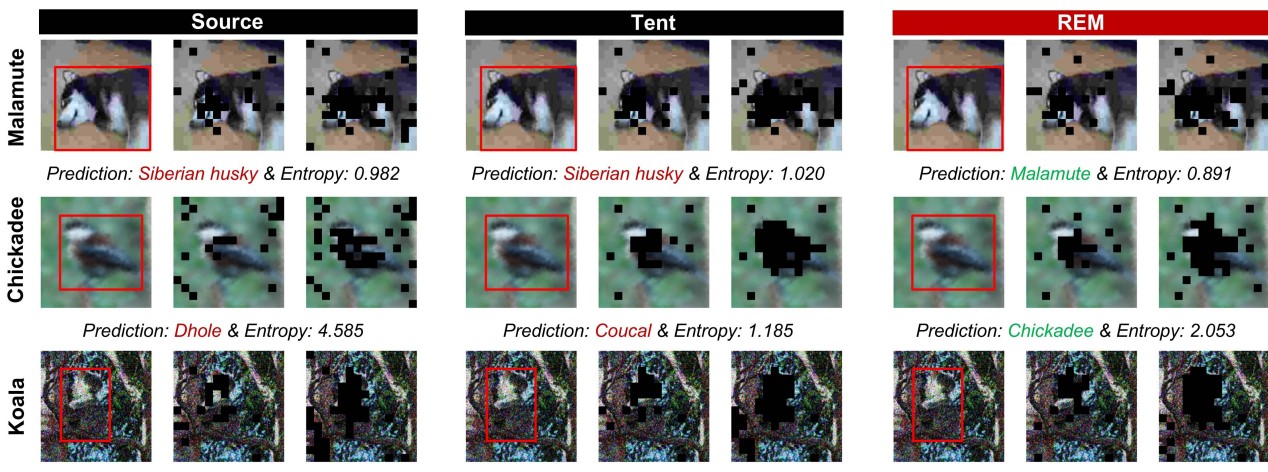

*Figure 7.* **Masked image visualization.** We compare the predictions and entropy of REM, Tent, and Source and visualize the results of our masking strategy. Each column represents images with masking ratios of 0, 10%, and 20% for each method, while each row shows the true label on the left, along with correct and incorrect predictions and its corresponding entropy values.

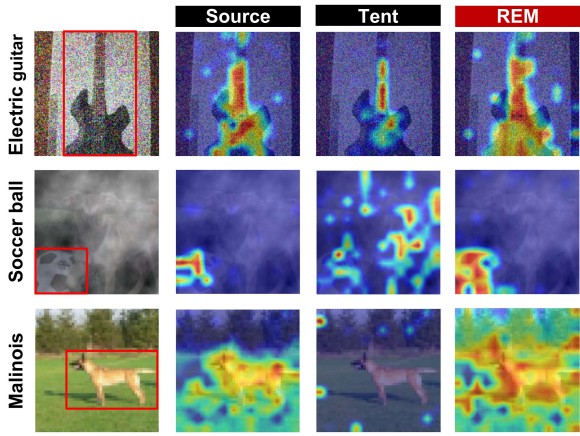

*Figure 8.* **Grad CAM visualization.** We compare attention maps to identify the pixels contributing to predictions.

*Table 5.* **Efficiency comparison.** We provide the number of training parameters, total time, number of the forward passes (FP) and models, and error rate (%).

| Method | Parameters↓ | Total Time↓ | # of FP | Total Models | Error |
|---|---|---|---|---|---|
| Source | 0 | - | 1 | $M_{src}$ | 55.8 |
| Tent | 0.04M | 8m35s | 1 | $M_{test}$ | 51.0 |
| CoTTA | 86.4M | 33m23s | 3or35 | $M_{test} + M_{ema} + M_{src}$ | 54.8 |
| ViDA | 93.7M | 54m48s | 12 | $M_{test} + M_{ema}$ | 43.4 |
| Continual-MAE | 86.5M | 59m56s | 12 | $M_{test} + M_{src}$ | 42.5 |
| REM | 0.03M | 17m21s | 3 | $M_{test}$ | 39.2 |

high learning rates achieves the best performance of 42.1% on Seen domains, while slower adaptation yields the best performance of 41.4% on Unseen domains. Such a trend is also observed in Figure 4, which presents the performance across all domains for the model trained on sequential all domains. Based on the adaptation order, rapid adaptation demonstrates low error on domains learned later, while slow adaptation achieves low error on initial domains.

**Discussion.** The preceding experiments indicate that achieving generalized performance across diverse domains does not necessarily guarantee optimal performance in CTTA. Distinctive advantage of TTA over domain generalization lies in its capability to perform domain-specific adaptation through online learning, which is critical for addressing

domain shifts effectively. Figure 5 presents a comparative analysis of adaptability under varying learning rates, while Figure 6 illustrates performance trends across a broad range of learning rate settings. The robustness of the proposed method across diverse learning rate boundaries underscores its practical utility, as it enables flexible adaptation speed selection tailored to specific application requirements.

### 4.3. Qualitative Results

**Masked Image Visualization.** From Figure 7, we provide a visualization of the masked images generated by the explicit mask chaining, along with the predictions and entropy values for the corresponding original images. It can be observed that masking is specifically applied to the pixels where objects are located. Moreover, compared to Tent, which tends to be overly confident in uncertain predictions, this approach maintains higher entropy, allowing for improvements in the incorrect predictions of the initial source model and effectively mitigating overconfidence.

**Class Attention Map Visualization.** In Figure 8, we present the Grad-CAM (Selvaraju et al., 2017) visualization results to highlight the salient pixels influencing the

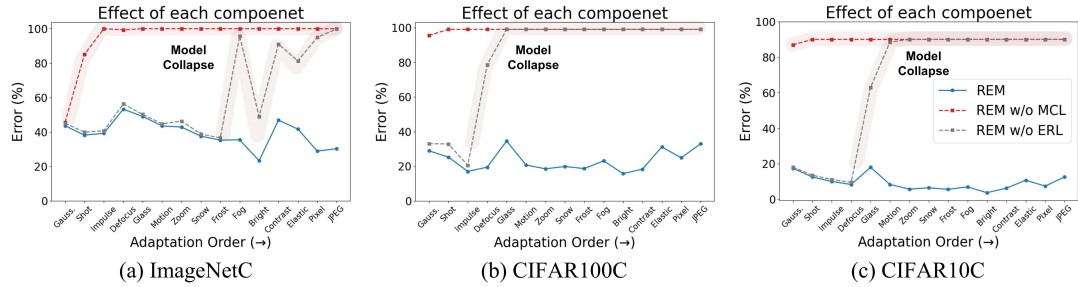

*Figure 9.* **Effect of each component.** We present the results of REM and compare them with variations where MCL and ERL are removed.

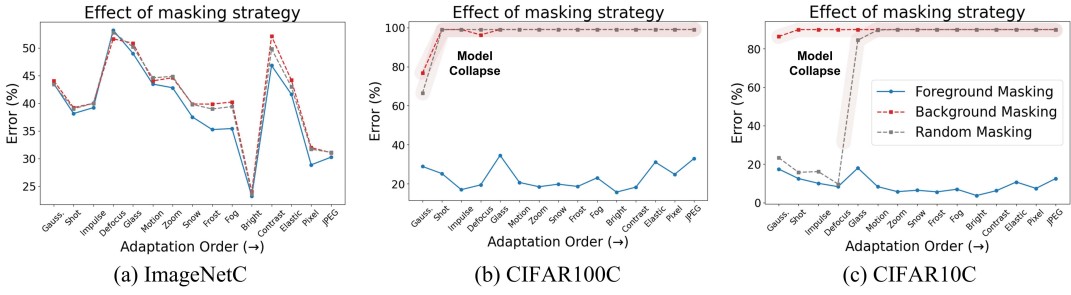

*Figure 10.* **Effect of masking strategy.** We present the results of REM (foreground masking) in comparison with different masking strategies, including background masking, and random masking.

model's predictions and to gain insights into its decision-making process. Each row corresponds to the results for Gaussian noise, fog, and pixelation, respectively, demonstrating that Tent gradually highlights local regions over time. This phenomenon can be interpreted as the model relying on non-discriminative local features rather than global semantic context for predictions, resulting in the generation of uniform predictions irrespective of the input. In contrast, our method adopts consistency regularization for masked objects and effectively captures comprehensive information related to context.

### 4.4. Efficiency Comparison

We compare the computational efficiency of our method with Tent, an EM approach, and CR based state-of-the-art methods such as CoTTA, ViDA, and Continual-MAE in Table 5. Our method follows the strategy of existing EM approaches that update only the normalization layers of a single test model ($M_{test}$), providing advantages in terms of training time, the number of trainable parameters, and the number of models that need to be stored. Recent CTTA approaches store the EMA model ($M_{ema}$) used as a teacher model and the source model ($M_{src}$) to regress to the initial source weights. In addition, it often requires numerous forward passes to model uncertainty. Compared to the recent state-of-the-art, Continual-MAE, we achieve a 3.3% performance improvement while requiring only 30% of the computation time and 0.03% of the training parameters.

### 4.5. Ablation Studies

**Effect of Each Component.** We present the results of the ablation experiments for each component in Figure 9. Model collapse appeared early for CIFARC, which contains lower resolution and information compared to ImageNetC, when MCL and REL were removed. Note that our method achieves stable performance without model collapse when both are applied, due to the organic design of each method.

**Effect of Masking Strategy.** In order to validate the appropriateness of our masking strategy, Figure 10 illustrates a comparison between our foreground masking strategy and cases involving background masking or random masking. Our intuition behind defining an explicit ranking relationship for the predictions is satisfied when masking the foreground. In this case, the method performed as designed, and there is no model collapse for all test datasets.

## 5. Limitation

Our study is grounded in the assumption that explicitly masking objects used as the basis for predictions can lead to a decrease in accuracy and an increase in entropy. While our method is simple and intuitive, it is not yet fully supported by rigorous theoretical proof. Despite efforts to address this, the counterexamples arising from the diversity of images still pose significant challenges. To mitigate this, we conduct several experiments that provide empirical evidence demonstrating statistical significance.

# 6. Conclusion

In this paper, we introduce Ranked Entropy Minimization (REM) to improve stability and efficiency in CTTA. Based on observation of model collapse, we propose a progressive masking strategy and dual complementary loss functions: masked consistency loss and ranked entropy loss. Consequently, REM captures the best of both worlds by integrating the stability of consistency regularization and the efficiency of entropy minimization. Through quantitative evaluations on various CTTA benchmarks, REM achieves state-of-the-art performance, demonstrating its effectiveness. Moreover, extensive qualitative experiments and ablation studies offer in-depth insights into the working principles. We hope that our work serves as a foundation for valuable discussions on computational cost in CTTA, paving the way for advances in efficiency and real-world applicability.

## Impact Statement

This paper presents work whose goal is to advance the field of Machine Learning. There are many potential societal consequences of our work, none which we feel must be specifically highlighted here.

## Acknowledgements

This work was supported by IITP grant funded by the Korea government(MSIT) (RS-2025-02283048, Developing the Next-Generation General AI with Reliability, Ethics, and Adaptability; RS-2023-00236245, Development of Perception/Planning AI SW for Seamless Autonomous Driving in Adverse Weather/Unstructured Environment; RS-2021-II212068, AI Innovation Hub) and NRF-2022R1A2C1091402.

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

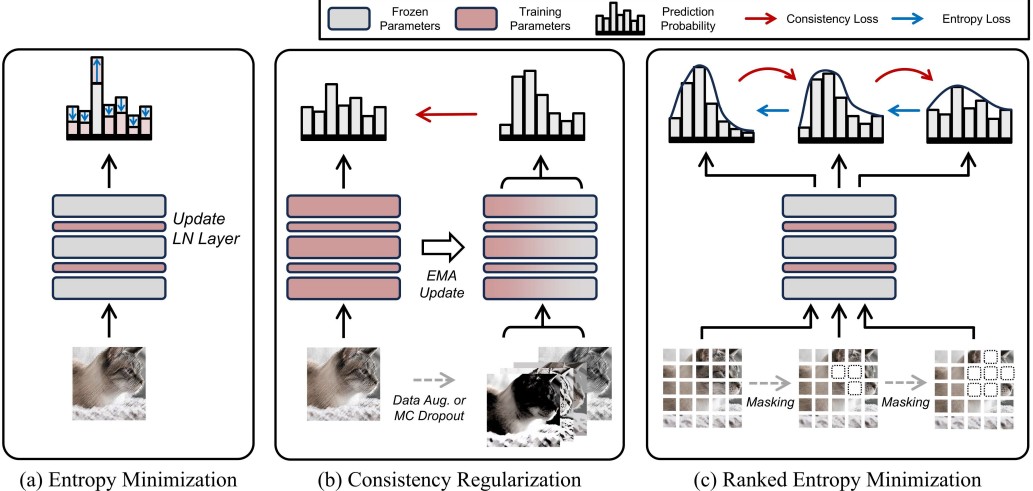

(a) Entropy Minimization       (b) Consistency Regularization       (c) Ranked Entropy Minimization

*Figure 11.* **Conceptual illustration comparing CTTA frameworks.** (a) Entropy minimization approach updates only the normalization layer for the target domain while utilizing a single model. (b) Consistency regularization method employs both a student model, which updates all parameters, and a teacher model, which is updated via exponential moving average (EMA). This approach enhances prediction diversity by numerous data augmentations or applying MC dropout to the model. (c) Our proposed method updates the normalization layer and adopts a single model. It integrates entropy minimization and consistency regularization with only two additional forward passes.

# A. Overall Framework and Comparison with Other Approaches

To improve clarity and facilitate comprehension, we present the overall frameworks of the Entropy Minimization (EM) approach, the Consistency Regularization (CR) approach, and our proposed method, REM in Figure 11. We maintain the training scheme of EM to preserve its computational efficiency. Additionally, we replace the traditional data augmentations used in CR approaches with explicit masking, eliminating the reliance on extensive image augmentations for enhancing predictive diversity. Consequently, our method preserves the computational efficiency of EM approaches while enhancing the robustness and generalization performance of CR approaches.

# B. Implementation Details

Table 6 provides details on the implementation of our experiments, including optimizer settings, learning rates, batch sizes, model architectures, and hyperparameters. For CTTA experiments, we follow the Continual-MAE framework. Specifically, for CIFAR datasets, we resize the input images to 384×384, while for all other experiments, the images are resized to 224×224.

*Table 6.* Implementation details.

| Task | CTTA | | TTA | | TTA-CLIP | |
|---|---|---|---|---|---|---|
| Dataset | ImageNetC | CIFARC | ImageNetC | ImageNet-R/V2/S | CIFAR | Other Datasets |
| **Experimental Protocols** | | | | | | |
| Reproducibility | Continual-MAE (Liu et al., 2024a) | | DeYO (Lee et al., 2024) | | WATT (Osowiechi et al., 2024) | |
| **Training Parameters** | | | | | | |
| Optimizer | Adam | Adam | SGD | SGD | Adam | Adam |
| Optimizer momentum | (0.9, 0.999) | (0.9, 0.999) | 0.9 | 0.9 | (0.9, 0.999) | (0.9, 0.999) |
| Learning rate | 1e-3 | 1e-3 | 1e-3 | 1e-3 | 1e-3 | 1e-4 |
| Batch size | 50 | 20 | 1 or 64 | 1 or 64 | 128 | 128 |
| Model architecture | ViT-B/16 | ViT-B/16 | ViT-B/16 | ViT-B/16 | CLIP-ViT-B/16 | CLIP-ViT-B/16 |
| **Algorithm Parameters** | | | | | | |
| $\lambda$ (Eq. 5) | 1.0 | 1.0 | 0.5 | 0.5 | 1.0 | 1.0 |
| m (Eq. 4) | 0 | 0 | 0 | 0 | 0 | 0 |

*Table 7.* Classification accuracy (%) on ImageNetC (severity level5) under online imbalanced label shifts (imbalance ratio = ∞).

| Label Shifts | Gaussian | shot | impulse | defocus | glass | motion | zoom | snow | frost | fog | brightness | contrast | elastic_trans | pixelate | jpeg | Mean↑ | Gain |
|---|---|---|---|---|---|---|---|---|---|---|---|---|---|---|---|---|---|
| Source (Dosovitskiy et al., 2021) | 9.4 | 6.7 | 8.3 | 29.1 | 23.4 | 34.0 | 27.0 | 15.8 | 26.3 | 47.4 | 54.7 | 43.9 | 30.5 | 44.5 | 47.6 | 29.9 | 0.0 |
| MEMO (Zhang et al., 2022) | 21.6 | 17.4 | 20.6 | 37.1 | 29.6 | 40.6 | 34.4 | 25.0 | 34.8 | 55.2 | 65.0 | 54.9 | 37.4 | 55.5 | 57.7 | 39.1 | 9.2 |
| Tent (Wang et al., 2021) | 53.1 | 53.1 | 54.3 | 54.2 | 51.5 | 58.6 | 52.4 | 3.5 | 7.8 | 69.5 | 74.8 | 67.0 | 58.7 | 69.2 | 66.2 | 52.9 | 23.0 |
| EATA (Niu et al., 2022) | 45.5 | 47.2 | 44.1 | 45.4 | 41.5 | 52.0 | 47.4 | 54.8 | 46.7 | 57.1 | 70.4 | 29.2 | 55.9 | 62.2 | 60.6 | 50.7 | 20.8 |
| SAR (Niu et al., 2023) | 53.1 | 53.3 | 54.3 | 54.0 | 52.1 | 58.0 | 52.7 | 8.6 | 28.6 | 69.1 | 74.7 | 66.7 | 59.1 | 67.1 | 64.9 | 54.4 | 24.5 |
| DeYO (Lee et al., 2024) | 52.9 | 54.8 | 55.4 | 54.1 | 55.6 | 62.1 | 34.4 | 64.6 | 63.7 | 71.1 | 77.1 | 64.2 | **67.2** | 72.4 | 68.2 | 61.2 | 31.3 |
| REM (Ours) | **57.0** | **57.2** | **58.1** | **58.6** | **56.3** | **63.2** | **58.4** | **67.3** | **67.5** | **74.4** | **78.9** | **70.5** | 65.6 | **73.3** | **70.0** | **65.1** | **35.2** |

*Table 8.* Classification accuracy (%) on ImageNetC (severity level5) under batch size 1.

| Batch Size 1 | Gaussian | shot | impulse | defocus | glass | motion | zoom | snow | frost | fog | brightness | contrast | elastic_trans | pixelate | jpeg | Mean↑ | Gain |
|---|---|---|---|---|---|---|---|---|---|---|---|---|---|---|---|---|---|
| Source (Dosovitskiy et al., 2021) | 9.5 | 6.8 | 8.2 | 29.0 | 23.5 | 33.9 | 27.1 | 15.9 | 26.5 | 47.2 | 54.7 | 44.1 | 30.5 | 44.5 | 47.8 | 29.9 | 0.0 |
| MEMO (Zhang et al., 2022) | 21.6 | 17.4 | 20.6 | 37.1 | 29.6 | 40.6 | 34.4 | 25.0 | 34.8 | 55.2 | 65.0 | 54.9 | 37.4 | 55.5 | 57.7 | 39.1 | 9.2 |
| Tent (Wang et al., 2021) | 52.1 | 51.8 | 53.2 | 52.4 | 48.7 | 56.5 | 49.5 | 8.5 | 15.2 | 67.3 | 73.4 | 66.7 | 52.6 | 64.9 | 64.3 | 51.8 | 21.9 |
| EATA (Niu et al., 2022) | 48.5 | 46.5 | 49.6 | 46.2 | 40.2 | 50.5 | 44.1 | 37.8 | 41.7 | 64.6 | 68.2 | 64.5 | 49.6 | 61.0 | 61.6 | 51.6 | 21.7 |
| SAR (Niu et al., 2023) | 52.0 | 51.7 | 53.1 | 51.7 | 48.9 | 56.8 | 50.6 | 16.8 | 54.8 | 67.2 | 74.7 | 66.1 | 55.3 | 66.8 | 65.2 | 55.5 | 25.6 |
| DeYO (Lee et al., 2024) | 54.6 | 55.6 | 56.0 | 55.5 | 17.3 | 62.7 | **59.5** | 65.6 | 64.4 | 72.0 | 77.3 | 10.9 | **66.3** | 71.8 | 68.7 | 57.2 | 27.3 |
| REM (Ours) | **57.4** | **57.8** | **58.6** | **59.2** | **56.9** | **63.5** | 59.1 | **68.4** | **67.5** | **74.5** | **79.0** | **71.1** | 65.7 | **73.3** | **70.5** | **65.5** | **35.6** |

*Table 9.* Classification accuracy (%) on ImageNetC (severity level 5 and level 3) under mixture of 15 corruption.

| Mixed Shifts | Level 5 | Level 3 |
|---|---|---|
| Source (Dosovitskiy et al., 2021) | 29.9 | 53.8 |
| Tent (Wang et al., 2021) | 24.1 | 70.2 |
| EATA (Niu et al., 2022) | 56.4 | 69.6 |
| SAR (Niu et al., 2023) | 57.1 | 70.7 |
| DeYO (Lee et al., 2024) | 59.4 | 72.1 |
| REM (Ours) | **62.4** | **74.0** |

*Table 10.* Classification accuracy (%) on ImagNet-R/V2/Sketch. Mean (%) denotes the average accuracy across 3 target domains.

| Domain Shifts | R | V2 | Sketch | Mean |
|---|---|---|---|---|
| Source (Dosovitskiy et al., 2021) | 59.5 | **75.4** | 44.9 | 59.9 |
| Tent (Wang et al., 2021) | 63.9 | 75.2 | 49.1 | 62.7 |
| CoTTA (Wang et al., 2022) | 63.5 | **75.4** | **50.0** | 62.9 |
| SAR (Niu et al., 2023) | 63.3 | 75.1 | 48.7 | 62.4 |
| FOA (Niu et al., 2024) | 63.8 | **75.4** | 49.9 | 63.0 |
| REM (Ours) | **64.3** | 75.2 | 49.7 | **63.1** |

## C. Experiments on Online Test-Time Adaptation Scenario

In addition to CTTA scenarios, our method is readily applicable to a wide range of TTA scenarios. To evaluate its effectiveness in a more challenging setting, we compare our approach against EM-based state-of-the-art methods, including MEMO (Zhang et al., 2022), Tent (Wang et al., 2021), EATA (Niu et al., 2022), SAR (Niu et al., 2023), and DeYO (Lee et al., 2024), in the wild online TTA scenarios proposed in SAR.

**Online Imbalanced Label Distribution Shifts.** Table 7 presents the performance comparison of TTA methods under class-imbalanced distributions across different domains. Our method achieves best performance across all domains except for elastic transform, improving the average performance by 3.9% compared to the previous state-of-the-art method, DeYO.

**Batch Size 1.** Table 8 shows the results for TTA under a batch size of 1, demonstrating the robustness of our method in scenarios where batch statistics cannot be effectively leveraged. Similar to the label shift scenario, our method achieves the best performance across all domains except for elastic transform, resulting in an 8.3% performance improvement.

**Mixed Distribution Shifts.** Table 9 presents the results for the TTA scenario where domain boundaries are ambiguous, leading to mixed domain distributions. Our method achieves performance improvements of 3.0% and 1.9% for corruption severity levels 5 and 3, respectively, indicating its potential for generalization across diverse domains.

**Domain Shifts.** Table 10 presents the TTA results for domain shifts from ImageNet to ImageNet-R (Hendrycks et al., 2021), ImageNet-V2 (Recht et al., 2019), and ImageNet-Sketch (Wang et al., 2019). We achieve a mean accuracy of 63.1% across all domains, surpassing the previous state-of-the-art FOA (Niu et al., 2024), which achieved 63.0%.

*Table 11.* Classification accuracy (%) comparison on vision-language model using CLIP ViT/B-16 across different datasets and domains.

| Dataset | Domain | CLIP | Tent | TPT | CLIPArTT | WATT | REM | REM+WATT |
|---------|--------|------|------|-----|----------|------|-----|----------|
| CIFAR | CIFAR-10 | 89.25 | 92.75 | 89.80 | 92.61 | 91.97 | $91.76_{\pm0.06}$ | $\mathbf{93.19}_{\pm0.14}$ |
| | CIFAR-100 | 64.76 | 71.73 | 67.15 | 71.34 | 72.85 | $69.15_{\pm0.05}$ | $\mathbf{72.87}_{\pm0.11}$ |
| | Mean | 77.01 | 82.24 | 78.48 | 81.98 | 82.41 | 80.46 | **83.03** |
| VisDA-C | 3D (trainset) | 87.16 | 87.57 | 84.04 | 87.58 | 87.72 | $87.45_{\pm0.00}$ | $\mathbf{88.95}_{\pm0.03}$ |
| | YT (valset) | 86.61 | 86.81 | 85.90 | 86.60 | 86.75 | $\mathbf{86.89}_{\pm0.01}$ | $86.84_{\pm0.03}$ |
| | Mean | 86.89 | 87.19 | 84.97 | 87.09 | 87.24 | 87.17 | **87.90** |
| Office-Home | Art | 79.30 | 79.26 | **81.97** | 79.34 | 80.43 | $80.17_{\pm0.11}$ | $80.29_{\pm0.11}$ |
| | Clipart | 65.15 | 65.64 | 67.01 | 65.69 | 68.26 | $66.96_{\pm0.04}$ | $\mathbf{68.32}_{\pm0.09}$ |
| | Product | 87.34 | 87.49 | **89.00** | 87.35 | 88.02 | $87.77_{\pm0.02}$ | $87.99_{\pm0.09}$ |
| | RealWorld | 89.31 | 89.50 | 89.66 | 89.29 | 90.14 | $\mathbf{90.14}_{\pm0.01}$ | $90.08_{\pm0.09}$ |
| | Mean | 80.28 | 80.47 | **81.91** | 80.42 | 81.71 | 81.26 | 81.67 |
| PACS | Art | 97.44 | 97.54 | 95.10 | 97.64 | 97.66 | $\mathbf{97.71}_{\pm0.00}$ | $97.64_{\pm0.06}$ |
| | Cartoon | 97.38 | 97.37 | 91.42 | 97.37 | 97.51 | $\mathbf{97.53}_{\pm0.00}$ | $97.45_{\pm0.02}$ |
| | Photo | **99.58** | **99.58** | 98.56 | **99.58** | **99.58** | $\mathbf{99.58}_{\pm0.00}$ | $\mathbf{99.58}_{\pm0.00}$ |
| | Sketch | 86.06 | 86.37 | 87.23 | 86.79 | 89.56 | $88.35_{\pm0.07}$ | $\mathbf{90.19}_{\pm0.14}$ |
| | Mean | 95.12 | 95.22 | 93.08 | 95.35 | 96.08 | 95.79 | **96.22** |
| VLCS | Caltech101 | **99.43** | **99.43** | 97.62 | **99.43** | 99.36 | $99.36_{\pm0.00}$ | $99.39_{\pm0.03}$ |
| | LabelMe | 67.75 | 67.31 | 49.77 | 67.74 | 68.59 | $68.06_{\pm0.12}$ | $\mathbf{69.26}_{\pm0.08}$ |
| | SUN09 | 71.74 | 71.57 | 71.56 | 71.67 | 75.16 | $75.04_{\pm0.04}$ | $\mathbf{75.76}_{\pm0.10}$ |
| | VOC2007 | 84.90 | **85.10** | 71.17 | 84.73 | 83.24 | $83.79_{\pm0.12}$ | $83.89_{\pm0.20}$ |
| | Mean | 80.96 | 80.85 | 72.53 | 80.89 | 81.59 | 81.56 | **82.08** |

# D. Experiments on Vision-Language Model

Our proposed REM can be applied in a plug-and-play manner and is adaptable to various modalities. We present experiments on the TTA setting with CLIP as the target model in Table 11. We compare the performance of our method against CLIP (Radford et al., 2021), Tent (Wang et al., 2021), TPT (Shu et al., 2022), CLIPArTT (Hakim et al., 2024), and WATT (Osowiechi et al., 2024). Following WATT, we report the TTA results from the CLIP model to CIFAR (Krizhevsky et al., 2009) and various domain adaptation and generalization benchmarks, including VisDA (Peng et al., 2018), Office-Home (Venkateswara et al., 2017), PACS (Li et al., 2017), and VLCS (Fang et al., 2013). As a result, our method achieves competitive performance compared to the previous state-of-the-art WATT, without requiring additional inner-loop training processes or ensemble methods. Furthermore, when WATT is combined with REM, it surpasses the existing results.

# E. Experiments on Practical TTA Scenarios

**Computational Time Constraint TTA Scenario.** Table 12 presents experimental results on ImageNet-3DCC (Kar et al., 2022) under the time-constrained protocol (Alfarra et al., 2024). We compare EATA (Niu et al., 2022) and our proposed method using ViT-B/16. EATA requires 2.41× the time relative to the adaptation speed of $g$ for which $C(g) = 1$, while REM requires 5.10× the time. Therefore, in the episodic scenario, where the model is re-initialized for each domain, REM shows lower performance than EATA due to its relatively slower adaptation. However, in the continual scenario, where domains are learned sequentially without model re-initialization, our method achieves higher performance due to the accumulation of learned knowledge and demonstrates stable adaptation across domains.

*Table 12.* Classification error rate (%) on ImageNet-3DCC under time constraint scenario

| Time | $t$ | | | | | | | | | | | | |
|------|------|------|------|------|------|------|------|------|------|------|------|------|------|
| Method | Depth of field | | Noise | | | Lighting | Weather | | Video | | | Camera motion | | Mean ↓ |
| | Near focus | Far focus | Color quant | ISO | Low light | Flash | Fog 3D | Bit err | H.265 ABR | H.265 CRF | XY-mot. blur | Z-mot. blur | |
| EATA-Episodic | 27.43 | 35.58 | 42.14 | 46.25 | 33.61 | 55.64 | 54.70 | 80.39 | 48.31 | 42.26 | 48.92 | 43.77 | **46.58** |
| REM-Episodic | 28.76 | 36.72 | 41.87 | 46.31 | 41.18 | 59.73 | 53.21 | 89.37 | 50.09 | 45.40 | 53.07 | 46.96 | 49.39 |
| EATA-Continual | 26.81 | 33.21 | 40.66 | 43.94 | 35.05 | 57.21 | 56.28 | 83.63 | 55.41 | 47.47 | 56.07 | 49.86 | 48.80 |
| REM-Continual | 29.30 | 33.95 | 41.12 | 43.65 | 31.85 | 56.24 | 51.20 | 87.17 | 49.07 | 41.38 | 49.35 | 43.33 | **46.47** |

# F. Experiments on Different Network Architectures

The proposed REM leverages the self-attention mechanism of ViT, yet it can be readily extended to other architectures as long as a ranked structure of difficulty can be explicitly defined through chained masking. To demonstrate generality, we introduce two additional variants: one based on feature activation (FA), where the attention map is computed as the average L2-norm of feature vectors across all spatial positions, and another based on Grad-CAM (Selvaraju et al., 2017) to modulate the masked regions. As shown in Table 13, FA-based REM consistently improves performance across various Transformer architectures. Additionally, Table 14 demonstrates its applicability to CNNs, where our method achieves notable gains even without the use of self-attention, highlighting the broad utility of the proposed difficulty-aware masking strategy.

*Table 13.* Mean error rate (%) on ImageNetC using transformer architectures

| Model | Source | Tent | CoTTA | ViDA | REM |
|---|---|---|---|---|---|
| Mobile-ViT-S | 75.28 | 75.61 | 75.72 | 75.27 | 74.28 |
| SwinTransformer-B | 59.26 | 73.17 | 46.84 | 57.84 | 46.56 |

*Table 14.* Mean error rate (%) on ImageNetC using CNN architectures

| Model | CoTTA | EATA | EcoTTA | BECoTTA | REM (FA) | REM (Grad-CAM) |
|---|---|---|---|---|---|---|
| WideResNet-28 | 16.2 | 18.6 | 16.8 | - | 16.9 | 16.5 |
| WideResNet-40 | - | 37.1 | 36.4 | 35.5 | 34.5 | 34.6 |

# G. Calibration Error Analysis

We investigate the mitigation of model collapse by analyzing the issue of overconfidence through model calibration error. We observe a consistent trend in which the Expected Calibration Error (ECE) tends to increase as the model outperforms the initial source model (Naeini et al., 2015). Notably, our method maintains a low ECE while achieving low error rates, emphasizing the practical importance of calibration for reliable and robust adaptation under distribution shifts.

*Table 15.* Comparison of ECE and error rates on ImageNetC.

| ImageNet-C | Source | Tent | SAR | ViDA | REM |
|---|---|---|---|---|---|
| ECE (%) $\downarrow$ | 5.3 | 12.6 | 10.3 | 14.6 | 8.7 |
| Error (%) $\downarrow$ | 55.8 | 51.0 | 45.2 | 43.4 | 39.2 |

# H. Comparison with Augmentation-based EM approaches

Recent augmentation-based EM methods (Marsden et al., 2024; Lee & Chang, 2024) adopt a variety of stochastic augmentations, such as color jitter and random affine transformations, similar to CoTTA and its variants. In contrast, our method introduces an interpretable and sample-specific augmentation scheme based on structured masking. By designing a ranked prediction distribution, we progressively refine the model's predictions while preserving the relative ranking, offering an intuitive and effective adaptation mechanism. ROID and CMT, like EATA, incorporate the Active Sample Criterion (ASC), which omits the backward pass for inaccurately predicted samples by setting their loss to zero. When ASC is applied to our method, it achieves a similar level of computational efficiency. As shown in Table 16, REM with ASC reduces inference time while maintaining competitive error rates. While ASC-based methods enable efficient adaptation by selectively updating on confident samples, our main objective is to leverage the entire set of test samples. Instead of discarding uncertain predictions, REM aims to reduce domain dependency at test time by enforcing a clear intra-image predictive structure, thereby enhancing robustness against unpredictable domain shifts.

*Table 16.* Comparison of total adaptation time and error rate on ImageNetC.

| Method | ROID | CMT | REM (N=1) | REM (N=1, ASC) |
|---|---|---|---|---|
| Time | 9m33s | 9m38s | 11m47s | 9m22s |
| Error (%) | 41.4 | 40.7 | 39.5 | 39.7 |

# I. Failure Case Analysis

We analyze the discrepancy between the outputs of original and masked images using Total Variation Distance (TVD) in Table 17, focusing on two domains where our method achieved significant performance improvements (Gaussian and Shot noise) and two domains where it showed relatively lower performance (Brightness and JPEG). Interestingly, the domains with successful performance gains, e.g., Gaussian and Shot noise, exhibited larger differences in the predicted probability distributions with and without masking. One possible interpretation is that, for relatively easier domains, a small discrepancy between the predicted distributions of the original and masked images may lead to a low loss, which in turn could reduce the adaptation speed. This observation suggests that dynamically adjusting the loss magnitude based on the estimated domain gap may further enhance the adaptation performance. In particular, incorporating an adaptive loss weighting scheme could help balance learning across domains of varying difficulty. We consider this a promising direction and leave its detailed exploration for future work.

*Table 17.* Total Variation Distance (TVD) for the first and last 50% samples under various corruptions on CIFAR100C.

| CIFAR100C | Gaussian | Shot | Brightness | Jpeg |
|---|---|---|---|---|
| TVD (first 50%) | $5.54 \pm 1.36$ | $3.44 \pm 1.38$ | $1.69 \pm 0.45$ | $2.90 \pm 0.95$ |
| TVD (last 50%) | $5.03 \pm 1.38$ | $3.82 \pm 0.69$ | $1.55 \pm 0.41$ | $2.59 \pm 0.78$ |

# J. Hyperparameter Sensitivity Analysis

REM incorporates three hyperparameters: the masking ratio $M_N$ in Equation (3), the margin m in Equation (4), and the weighting coefficient $\lambda$ in Equation (5). We provide an ablation study on the masking ratio in Table 18. Although the best performance is achieved when $N = 3$, we adopt the combination $M_N = \{0, 5\%, 10\%\}$ to strike a balance between computational complexity and accuracy. Moreover, Figure 12 presents the results of a grid search over all hyperparameters, reporting the mean error across all domains in the ImageNet-to-ImageNetC benchmark. The experimental results indicate that our method exhibits low sensitivity to hyperparameter variations. Based on these results, we set $\lambda = 1$ and m $= 0$ for the CTTA experiments.

*Table 18.* Mean error rate (%) for different masking ratios $M_N$ on ImageNetC

| $M_N$ | $M_1$ | $M_2$ | $M_3$ |
|---|---|---|---|
| +5% | $\{0, 5\%\}$ | $\{0, 5\%, 10\%\}$ | $\{0, 5\%, 10\%, 15\%\}$ |
| Error | 40.6% | 39.4% | 38.9% |
| +10% | $\{0, 10\%\}$ | $\{0, 10\%, 20\%\}$ | $\{0, 10\%, 20\%, 30\%\}$ |
| Error | 39.7% | 39.2% | 39.4% |
| +15% | $\{0, 15\%\}$ | $\{0, 15\%, 30\%\}$ | $\{0, 15\%, 30\%, 45\%\}$ |
| Error | 39.5% | 39.4% | 40.0% |

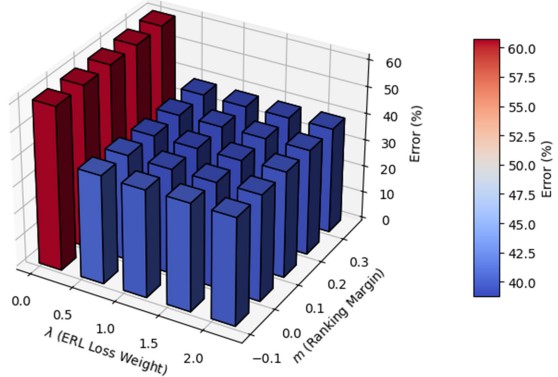

*Figure 12.* Hyperparameter sensitivity analysis

