# OpenReview forum: "Ranked Entropy Minimization for Continual Test-Time Adaptation"
_ICML.cc/2025/Conference — ICML 2025 poster_

### Official Review · Reviewer_VSix · 2025-03-11

**Overall Recommendation:** 4

**Summary:**

This work studies continual test-time adaptation and proposes a method to tackle it. In particular, based on experimental motivation on how entropy minimization collapses under continual TTA to predicting the same class, they propose to optimize the cross entropy between the prediction of the model on two masked versions of the input. Experiments are conducted on ImageNet-C, CIFAR10/100-C showing consistent performance gain under the provided baselines.

**Claims And Evidence:**

While the provided experiments generally support the claims of this work, the only claim that is not fully supported is the efficiency claim. The summation in Equation (3) and (4) requires 2*N forward passes. This makes the adaptation either very slow (under fixed computational budget), or very computational intensive. Further, it is Also not clear how costly it is to construct the N masks for each received input image at test time. While Table 5 shows that REM requires only 3 forward passes, extra discussion on the computational requirement of REM is necessary.

**Essential References Not Discussed:**

- 3D Common Corruptions for Object Recognition, CVPR 2022.

**Experimental Designs Or Analyses:**

While I really like the experimental analysis presented in this work (especially the ones in Figure 3), certain experiments are missing and it would make the paper much stronger to include them:

(1) Missing baselines: There are a couple of strong missing baselines in the comparisons tables such as EATA  and SAR. It is important to compare against both method in the main paper.

(2) Missing ablation: There are two main missing ablations in this work: analyzing the impact $\lambda$ and the impact of $M$.

(3) Computational budgeted evaluation: Since the proposed method requires additional forward passes making it more computationally intensive, it is important to compare different methods (especially the efficient EATA) under computational time constraint settings [B].

(4) Evaluation schemes: while this work mainly focuses on continual TTA, it is important to show experiments under different evaluation protocols; namely practical TTA [C].

[B] 3D Common Corruptions for Object Recognition, ICML 2024.

[C] Robust Test-Time Adaptation in Dynamic Scenarios, CVPR 2023.

**Methods And Evaluation Criteria:**

While the proposed method is evaluated on ImageNet-C and CIFAR10/100-C, it is also important to report the results on the more realistic benchmark ImageNet-3DCC [A]

[A] 3D Common Corruptions for Object Recognition, CVPR 2022.

**Other Comments Or Suggestions:**

There are a couple of typos in that paper:

1) In line 325-326: " Section 4.2 presents the performance on the 5 unseen domains after training on 10 domains." I think here you should refer to the Table, rather than referring to the section you are at.

2) In line 418-419: "Compared to the re- cent state-of-the-art, Continual-MAE,", I believe the first comma should be replaced with ;.

**Other Strengths And Weaknesses:**

Please refer to the earlier sections of my review.

**Questions For Authors:**

In addition, I have the following question to be clarified:

- How are the supervised learning results obtained? As far as I know, ImageNet-C does not have any training sets.

**Relation To Broader Scientific Literature:**

The findings of this work are related to the TTA literature

**Theoretical Claims:**

There are not theoretical claim in this work

---

> ### Author Rebuttal · Authors · 2025-03-30
>
> Thank you for your positive feedback and insightful suggestions. Below, we provide detailed responses to your questions.
>
> >**1. Computational intensive for mask**
>
> A1. The token-wise attention used to compute the mask is applied only to the final self-attention layer and accounts for a very small portion of the overall forward pass. We provide an analysis of the mask generation time  per batch with respect to $N$ in the table below:
> Forward pass|Mask computation ($N=1$)|Mask computation ($N=2$)
> :---:|:---:|:---:
> 7.012$\pm$0.390ms|0.091$\pm$0.008ms|0.134$\pm$0.007ms
>
> Although our method relatively requires more total time than entropy minimization methods, it achieves higher accuracy with lower computational cost compared to recent state-of-the-art CTTA methods based on consistency regularization. Our goal is to find a compromise between the efficiency of entropy minimization and the stability of consistency regularization methods. Positioned in between these two paradigms, our method achieves the highest classification performance with balanced efficiency.
>
> >**2. Additional baseline**
>
> A2. Thanks to your suggestion, we provide the following table comparing our method with the additional baselines:
> Method|EATA|SAR|Ours
> ---|---|---|---
> ImageNetC|41.3|45.2|**39.2**
>
> >**3. Ablation for mask parameter**
>
> A3. We added an ablation study on masking. As a trade-off between computational efficiency and accuracy, we utilize masks with ratios of {0, 10%, 20%}, and the results regarding the lambda parameter are provided in Appendix E.
> M_N (N=1)|{0,5%}|{0,10%}|{0,15%}
> ---|---|---|---
> Error|40.6|39.7|39.5
> M_N (N=2)|{0,5%,10%}|{0,10%,20%}|{0,15%,30%}
> Error|39.4|39.2|39.4
> M_N (N=3)|{0,5%,10%,15%}|{0,10%,20%,30%}|{0,15%,30%,45%}
> Error|38.9|39.4|40.0
>
> >**4. Experiment on realistic TTA**
>
> A4. Thank you for suggesting an evaluation under realistic scenarios to verify the practical applicability of our method. The table below presents the experimental results on the ImageNet-3DCC [A] dataset under the time-constrained protocol [B]. We compare EATA and our proposed method using ViT-B/16. EATA requires 2.41$\times$ the time when $C(g)=1$, while REM requires 5.10$\times$ the time. Therefore, in the *episodic* scenario, where the model is re-initialized for each domain, REM shows lower performance than EATA due to relatively slower adaptation. However, in the *continual* scenario, where domains are learned sequentially without model re-initialization, our method achieves higher performance due to the accumulation of learned knowledge and demonstrates stable adaptation across domains.
> ImageNet-3DCC|Depth of field|Noise|Lighting|Weather|Video|Camera motion
> :---:|:---:|:---:|:---:|:---:|:---:|:---:
> EATA-Episodic|31.5|40.7|55.6|54.7|57.0|46.3
> Ours-Episodic|32.7|43.1|59.7|53.2|61.6|50.0
> EATA-Continual|30.0|39.9|57.2|56.3|62.2|53.0
> Ours-Continual|31.6|38.9|56.2|51.2|59.2|46.3
>
> [A] 3D Common Corruptions for Object Recognition, CVPR 2022.
> [B] Evaluation of Test-Time Adaptation Under Computational Time Constraints, ICML2024.
>
> >**5. Practical TTA evaluation**
>
> A5. We also present experiments on CIFAR10C under practical TTA protocol [C] in the following table. These results confirm that our method operates robustly across various realistic scenarios as suggested.
> CIFAR10C|CoTTA|ViDA|Continual-MAE|Ours
> ---|---|---|:---:|---
> Error(%)|79.2|27.4|15.7|**14.2**
>
> [C] Robust Test-Time Adaptation in Dynamic Scenarios, CVPR 2023.
>
> >**6. Details in supervised learning**
>
> A6. The supervised result refers to the outcome obtained by training on the test set using cross-entropy loss with access to target labels in an online manner. Since target labels are not accessible during test time in TTA, this serves as an upper bound of the adaptation performance.
>
> >**Minor**
>
> Thank you for pointing out the typo. We have corrected the reference from Sec 4.2 to Table 4 and fixed the typo accordingly.

---

> > ### Comment · Reviewer_VSix · 2025-04-08
> >
> > I would like to thank the authors for the efforts put in responding to my comments. Thus, I am raising my score to 4.

---

### Official Review · Reviewer_wsjg · 2025-03-13

**Overall Recommendation:** 2

**Summary:**

This paper addresses the model collapse issue in CTTA. Specifically, it aims to reconcile the trade-off between fast but unstable EM methods and stable yet computationally expensive CR methods. The authors propose REM, a novel EM-based approach incorporating a progressive masking strategy. This strategy gradually adjusts prediction difficulty by structuring entropy sequentially, thereby aiming to enhance both stability and adaptability.

## update after rebuttal
The masking augmentation strategy proposed by the authors seems reasonable and well-motivated. That said, this approach relies on the assumption that ranking should be preserved, which can also be evaluated using more standard augmentation techniques such as varying rotation angles or crop levels. Given this, it is challenging to attribute the observed effects specifically to the proposed ranking structure. Furthermore, the performance gains over prior methods that combine entropy loss with data augmentation are relatively modest. For these reasons, I would like to retain my original score.

**Claims And Evidence:**

**Strengths:**

- Clearly visualizes the model collapse phenomenon and convincingly demonstrates the effectiveness of the proposed REM method through informative graphs and illustrations.

**Weaknesses:**

- The trivial solution-induced model collapse problem in EM-based methods has already been analyzed extensively in prior works such as EATA, SAR, and DeYO. This diminishes the novelty of the analytical contribution in Section 2.
- The paper lacks a comparative analysis between the proposed masking-based augmentations and diverse existing augmentation techniques, thus insufficiently supporting the claimed novelty.

**Essential References Not Discussed:**

I keep up with the literature in this area.

**Experimental Designs Or Analyses:**

**Strengths:**

- The effectiveness of various masking strategies (Fig. 10) and the impact of removing specific loss terms (Fig. 9) were thoroughly analyzed, experimentally demonstrating that MCL and ERL significantly contribute to preventing model collapse.

**Weaknesses:**

- Lack of evaluation regarding the robustness of the proposed object-focused masking strategy in highly domain-shifted scenarios. Additional experiments or analyses are necessary to clearly establish its effectiveness under extreme domain shifts.
- Inadequate analysis of failure cases. The manuscript does not provide an experimental investigation into situations where REM fails, which is essential for a comprehensive understanding of method limitations.
- A comparison with EM approaches using varying intensities of augmentation is necessary to demonstrate the relative advantage of the proposed masking augmentation more explicitly.
- Although Appendix C reports accuracy values, the use of downward arrows in the tables introduces confusion and could mislead readers regarding the interpretation of experimental results. Clarifying the representation method would improve the manuscript's readability.

**Methods And Evaluation Criteria:**

**Strengths:**

- Employs standard benchmarks for CTTA evaluation and widely-adopted ViT models, thereby ensuring credibility and comparability of results.

**Weaknesses:**

- Although REM claims a novel mechanism to prevent model collapse, its fundamental idea of diverse augmentations for entropy minimization has already been explored in existing literature [a,b].

> [a] Marsden et al., "Universal test-time adaptation through weight ensembling, diversity weighting, and prior correction," WACV, 2024.
>

> [b] Lee & Chang, "Continual momentum filtering on parameter space for online test-time adaptation," ICLR, 2024.
>
- Table 5 reveals that REM requires more than ten times the computational resources compared to TENT, questioning its practical viability.

**Other Comments Or Suggestions:**

see above

**Other Strengths And Weaknesses:**

see above

**Questions For Authors:**

see above

**Relation To Broader Scientific Literature:**

see above

**Theoretical Claims:**

**Weaknesses:**

- The theoretical explanation supporting REM's capability to prevent model collapse, as mentioned in Section 5, is insufficiently developed.
- The paper lacks a clear logical analysis explaining why the proposed progressive masking strategy prevents model collapse.
- Although the authors assert that sequential entropy structuring avoids overconfidence in difficult samples, they do not rigorously analyze how sequential entropy ordering directly mitigates model collapse.

---

> ### Author Rebuttal · Authors · 2025-03-31
>
> We appreciate your constructive suggestions. We address your questions below.
> >**1. Model collapse problem**
>
> Model collapse in entropy minimization methods is a well-known issue, but we would like to emphasize that it remains an unresolved challenge in achieving stability during TTA process. Fig. 2 shows the sensitivity of entropy minimization to training strategy (learning rate), suggesting that even small variations can lead to model collapse in TTA. Fig. 6 also shows the model’s sensitivity to various learning rates. Since entropy minimization operates within a narrow range where collapse does not occur, it requires careful tuning. In contrast, our proposed method shows stable adaptation across different training strategies highlighting its relative robustness against model collapse, and we provide the comparison table with prior works.
> Method|EATA|SAR|DeYO|Ours
> ---|---|---|---|---
> ImageNetC|41.3|45.2|42.9|**39.2**
>
> >**2. Explicit mask chaining augmentation**
>
> Augmentation-based EM methods[a,b], which employ diverse data augmentations, apply stochastic augmentations, e.g., color jitter and random affine, similar to CoTTA and its variants. In contrast, our method is distinguished by applying interpretable, sample-specific, masking-based data augmentation. By designing a ranked prediction distribution, we offer an intuitive and simple solution that progressively refines the prediction distribution while preserving the ranking order. We provide the comparison table with [a,b].
> Method|ROID[a]|CMF[b]|Ours
> ---|---|---|---
> ImageNetC|41.4|40.7|**39.2**
>
> >**3. Computational resource compared with TENT**
>
> In Table5, our method requires approximately twice the training time (_not ten times_, tent: 8min vs ours: 17min) compared to TENT; however, it achieves higher performance while reducing the computational cost to 1/3 compared to Continual-MAE (59min). EM methods are efficient but raise concerns about model collapse, whereas mean teacher-based methods offer stability at the cost of high computational overhead. Our goal is to integrate the strengths of both approaches to achieve a balanced solution from the viewpoint of both computational efficiency and classification performance.
>
> >**4. Theoretical explanation**
>
> While our method provides a solution to mitigate model collapse, it does not theoretically guarantee complete prevention. Therefore, we conducted extensive experiments to empirically validate the effectiveness of our approach. In Fig. 3, the analysis of entropy and accuracy with respect to different masking ratios supports the soundness of our method’s intuition. The visualizations of masked images in Fig. 7 show that masking is applied as intended. Lastly, Fig. 8 shows class attention map visualizations, revealing that entropy minimization methods often rely on class-irrelevant local pixels as the basis for predictions, leading to suboptimal solutions.
>
> >**5. Model collapse**
>
> We observed that entropy minimization can lead to predictions collapsing into a single class, regardless of the input. To mitigate this issue, it is important to focus on contextual information as the basis for prediction while addressing the influence of incorrect supervision signals. Our method alleviates model collapse by improving relative relationships rather than relying on direct supervision signals, through relational algorithms: our MCL, which focuses on contextual information, and our ERL, which preserves the ranking order of entropy based on prediction difficulty.
>
> >**6. Avoid overconfidence**
>
> In Reviewer MEbz’s A5, we investigate the effect of the margin in the entropy ranking loss. As the margin increases, the influence on the ranking diminishes, leading to a rise in ECE. This suggests that as the weight on direct entropy minimization decreases, the overconfidence issue is alleviated.
>
> >**7.  Highly domain-shifted scenario**
>
> In addition to the CTTA scenario of the main paper, we conducted experiments under standard protocols for various domain shifts in Appendix C and D, including Online TTA and Vision-Language Model(VLM)-based TTA scenarios. Our method shows successful adaptation not only on datasets where corruptions are applied to the source domain but also in cases involving substantial domain shifts, e.g., adapting from ImageNet to ImageNet-R, ImageNet-V2, and ImageNet-Sketch. We also validate ours using a realistic benchmark with ImageNet-3DCC (please, refer to Reviewer VSix’s A4).
>
> >**8. Failure case**
>
> We establish a ranking structure using 10% and 20% masked images. However, when the object is entirely masked, there is a possibility that some predictions may become biased toward the background, as shown in the bottom-right visualization of Fig. 8. To alleviate this issue, we propose Entropy Ranking Loss in Sec. 3.4. Moreover, please refer to reviewer sBwW’s A2 regarding the failure case related to distributional discrepancy analysis.
>
> >**Minor**
>
> We revised the arrow in Appendix C to point upward.

---

> > ### Comment · Reviewer_wsjg · 2025-04-04
> >
> > Thank you for the authors' response and the additional experiments. However, some questions remain.
> >
> > - As with the proposed progressive masking method, it seems possible to design an entropy ranking loss by adjusting the intensity of various existing augmentation techniques. This experiment could highlight a distinction from existing methods that combine augmentations with EM. Does this kind of experimental design also lead to performance improvements?
> > - The table shows a slight performance improvement compared to the existing methods (i.e., ROID, CMF) that combine augmentation and EM. How does the performance relative to computational cost compare between the existing methods and REM?
> > - Even if there is no theoretical guarantee, could you describe the theoretical basis for why this method works well?

---

> > > ### Author Response · Authors · 2025-04-06
> > >
> > > Thanks for your constructive suggestions. We leave the following responses for solving your concerns.
> > >
> > > >**A1. Augmentation methods**
> > >
> > > Yes, we agree that other solutions could work, provided that they can ensure a structured ranking relationship for explicit mask chaining. However, we would like to emphasize that our foreground masking strategy offers an intuitive solution by reducing randomness, thereby helping to preserve this ranking structure. To address your concern, we provide the following simple extension experiments using the original-weak-strong augmentation strategy.
> > >
> > > ImageNetC|Noise|Blur|Weather|Digital
> > > ---|---|---|---|---
> > > RandAug|43.0|48.1|33.8|78.3
> > > Ours|40.3|47.1|32.8|36.9
> > >
> > > Since our loss function is based on the ranking relationship between accuracy and entropy, we observe that when this ranking is preserved, the existing augmention methods works. However, when the ranking is not preserved, it fails to mitigate collapse depending on the adaptation order (e.g., from Noise to Digital), partially due to the failure in selecting an appropriate augmentation strategy. It can also be interpreted as evidence of the robustness of our method, which provides a simple and intuitive way to ensure a clear prediction ranking structure.
> > >
> > > >**A2. Computational cost comparison**
> > >
> > > ROID and CMT, like EATA, employ the Active Sample Criterion (ASC), which sets the loss of inaccurate samples to zero, thereby skipping the backward pass for certain samples. Therefore, when $N=1$, applying ASC to our method can achieve a similar level of computational cost. ASC that utilizes only accurate samples achieves stable adaptation and is a promising approach for achieving efficiency by not training on all data. However, as explained in our response to reviewer f4Hw under "Comparison with entropy-based method (A6)," our major goal is on the method that leverages the entire set of samples and we aim to reduce the domain-dependency at test time by establishing clear predictive relationship within the image, thereby mitigating the impact of unpredictable domain shifts.
> > >
> > > Method|ROID|CMT|Ours (N=1)|Ours (N=1, ASC)
> > > ---|---|---|---|---
> > > Time|9m 33s|9m 38s|11m 47s|9m 22s
> > > Error|41.4|40.7|39.5|39.7
> > >
> > > >**A3. Theoretical basis**
> > >
> > > We appreciate the reviewer’s insightful comments regarding the theoretical foundation of our work. We would like to clarify that our approach is primarily grounded in empirical observations rather than rigorous theoretical analysis (as provided in authors’ rebuttal A5). Nonetheless, in an effort to offer a theoretical perspective, we seek to draw a connection between types of errors using the Bayes risk within the probably approximately correct (PAC) learning framework [R41]. In particular, [R42] demonstrates that for two hypothesis spaces, $\mathcal{F}_1 \subseteq \{f: \mathcal{X} \rightarrow \mathcal{Y}\}$ and $\mathcal{F}_2 \subseteq \{f: \mathcal{X} \rightarrow \mathcal{Y}\}$, if $\mathcal{F}_1 \subseteq \mathcal{F}_2$, then the approximation errors satisfy:
> > >
> > > $Err_D^{apx}(\mathcal{F}_1) \geq Err_D^{apx}(\mathcal{F}_2)$.
> > >
> > > Inspired by these results, we try to conceptually model the relationship between errors under explicit augmentations. However, this requires a strong assumption that the information contained in the augmented data is less than or equal to that of the original data, which is a condition that is not always guaranteed in the wild. To address this challenge, we proposed a simple yet intuitive solution: reducing the information content via foreground masking, thereby aligning with the assumption in a more controlled manner.
> > >
> > > In the context of our method, $\mathcal{F}_1$ corresponds to the model incorporating masking, while $\mathcal{F}_2$ represents the model without it. Our explicit mask chaining mechanism can be interpreted as an extension to a broader hypothesis space $\mathcal{F}_N$.
> > >
> > > By reducing the approximation error $Err_D^{apx}(\mathcal{F}_1)$ associated with the masked prediction, the proposed MCL improves the upper bound of the original prediction’s approximation error $Err_D^{apx}(\mathcal{F}_2)$, thereby mitigating model collapse and promoting progressive performance improvement. Furthermore, our ERL is designed to maintain a consistent ranking structure between $\mathcal{F}_1$ and $\mathcal{F}_2$.
> > >
> > > While this perspective aligns with our intuition, it has not been rigorously proven. Out of consideration for the potential impact on the ML community, we chose not to include this theoretical reasoning in the main paper. We are transparent in stating this limitation in Section 5, noting the absence of a formal theoretical justification.
> > > Nonetheless, we hope that this supplementary explanation of the theoretical concept behind our work helps to address the reviewer’s concerns regarding the theoretical basis.
> > >
> > > [R41] Estimating the bayes risk from sample data, NeurIPS1995.
> > > [R42] Sample-specific Masks for Visual Reprogramming-based Prompting, ICML2024.

---

### Official Review · Reviewer_sBwW · 2025-03-13

**Overall Recommendation:** 4

**Summary:**

This paper proposes a novel Ranked Entropy Minimization method for test-time adaptation. While leveraging entropy as a supervision signal may risk model prediction collapse, the authors address this challenge by first constructing an explicit masking chain with varying masking ratios on the original images. They introduce two critical constraints: enforcing consistency between predictions of highly masked images and those with lower masking ratios, and ensuring that the entropy of predictions for low-masking-ratio images remains consistently lower than that of high-masking-ratio images. These constraints effectively enhance the model’s cross-domain generalization capability while updating only a small number of parameters. The authors validate the effectiveness of their approach through comprehensive experiments across multiple benchmarks, demonstrating significant improvements in stability and adaptability under continual test-time adaptation scenarios.

**Claims And Evidence:**

The paper presents a clear and compelling argument. The authors observe that relying solely on entropy minimization as a supervisory signal can lead to model collapse in certain scenarios due to convergence to trivial solutions, where probability distributions collapse to a single point in polar coordinate representations. Inspired by Zeno’s paradox of Achilles and the tortoise, the authors propose a novel method based on entropy constraints for masked images to address this issue. The study is strengthened by extensive visualizations and insightful observations that illustrate the research motivation and theoretical foundations.

**Essential References Not Discussed:**

The references cited in this paper are sufficient.

**Experimental Designs Or Analyses:**

This paper thoroughly validates the effectiveness of Ranked Entropy Minimization in continuous test-time adaptation tasks through comprehensive experiments, demonstrating significant improvements over existing methods on benchmark datasets such as ImageNet-C, CIFAR10-C, and CIFAR100-C. The visualization analysis reveals REM’s core mechanism: the self-attention-based foreground masking strategy precisely localizes object regions, forcing the model to learn global semantics through mask consistency loss and thereby avoiding model collapse caused by over-reliance on local features.

**Methods And Evaluation Criteria:**

This paper tightly integrates observations with the motivation to propose an insightful and effective algorithm. By constructing two complementary constraints, the authors successfully address the challenge of models collapsing into trivial solutions when relying solely on entropy minimization. The introduction of masking on critical regions of images generating high-entropy data while imposing constraints, which ensures the preservation of global semantic comprehension capabilities. Furthermore, visualizations of the masking strategy demonstrate that obscuring object-related regions effectively establish a hierarchy of prediction difficulty, thereby preventing model collapse.

**Other Comments Or Suggestions:**

None.

**Other Strengths And Weaknesses:**

This paper identifies a critical issue in test-time adaptation through a fascinating observational experiment: how to avoid model collapse caused solely by entropy constraints. The authors ingeniously propose two constraints: ensuring that the entropy of highly masked images is greater than that of lightly masked ones, and guiding the probability distribution of highly masked images to align with that of lightly masked images. These constraints effectively prevent prediction collapse. The motivation of the paper is clear, and the proposed method is highly effective.

**Questions For Authors:**

1. What are the advantages of constructing challenging samples through image masking in this paper compared to selecting challenging samples from a queue using active learning?

2. In Table 3, some experimental results show significant differences compared to the current state-of-the-art (SOTA) methods (e.g., for brightness and JPEG). For certain types of image corruptions, if the masking ratio is too high, could it lead to a situation where the output probability distribution of high-masking-ratio samples consistently fails to align with that of low-masking-ratio samples? I believe this analysis is crucial, as it is directly related to the masking ratio set during Explicit Mask Chaining.

**Relation To Broader Scientific Literature:**

[1] published at ICML 2024, primarily investigates the application of active learning in test-time adaptation. Both this study and the current paper highlight the positive role of challenging samples in enhancing model domain generalization and propose insightful methods from different perspectives.

[1]. Gui S, Li X, Ji S. Active Test-Time Adaptation: Theoretical Analyses and An Algorithm[C]//The Twelfth International Conference on Learning Representations.

**Theoretical Claims:**

I have reviewed the authors' formulation of Explicit Mask Chaining and the two constraints, and they align with the theoretical descriptions provided by the authors.

---

> ### Author Rebuttal · Authors · 2025-03-29
>
> We appreciate your constructive comments and positive reviews. We address your questions below in detail.
>
> >**1. Sampling through image masking vs. sampling from a queue using active learning**
>
> A1. The active approach of capturing accurate samples and adapting using selected samples is meaningful in that it prevents the model from being degraded by uncertain samples. However, in this paper, we are concerned that sample selection based on active learning may heavily depend on the initial model and may be difficult to generalize due to the need to define an entropy threshold against varying training environments. Instead, we explicitly control the learning difficulty while training on all samples equally, aiming for stable operation and maintaining the goal of continual adaptation.
>
> >**2. Failure case analysis**
>
> A2. Thanks to your insightful suggestion, we analyzed the discrepancy between the output of original and masked images using Total Variation Distance (TVD) for two domains where our method achieved significant performance improvements (Gaussian and Shot noise) and two domains where it showed relatively lower performance (Brightness and JPEG). Interestingly, the domains with successful performance gains, e.g., Gaussian and Shot noise, exhibited larger differences in the predicted probability distributions with and without masking. One possible interpretation is that, for relatively easier domains, a small discrepancy between the predicted distributions of the original and masked images may lead to a low loss, which in turn could reduce the adaptation speed. This led us to the insight that adjusting the loss magnitude according to the domain gap may further aid adaptation. We sincerely appreciate your suggestion, which enabled a deeper understanding of both the strengths and limitations of our method.
> CIFAR100C|Gaussian|Shot|Brightness|Jpeg
> ---|:---:|:---:|:---:|:---:
> TVD (first 50%)|5.54$\pm$1.36|3.44$\pm$1.38|1.69$\pm$0.45|2.90$\pm$0.95
> TVD (last 50%)|5.03$\pm$1.38|3.82$\pm$0.69|1.55$\pm$0.41|2.59$\pm$0.78

---

### Official Review · Reviewer_MEbz · 2025-03-13

**Overall Recommendation:** 3

**Summary:**

This paper proposes a masked consistency loss (MCL) and entropy ranked loss (ERL) based learning mechanism for continual test-time adaptation (CTTA).
The MCL involves incrementally masks the images for data augmentation, and the ERL contributes in ensuring that the entropy of predictions with a low masking ratio is lower than those with a high masking ratio.
The proposed approach utilizes the self-attention structure to cluster similar content in order to mask the content.
Experiments on ImageNet-to-ImageNetC, CIFAR10-to-CIFAR10C, and CIFAR100-to-CIFAR100C benchmarks using ViT-B/16 architecture.

**Claims And Evidence:**

Some of the claims are not well supported or lack evidence. Refer to Weaknesses and Questions.

**Essential References Not Discussed:**

Recent state-of-the-art methods are not discussed, as well as the comparisons are missing.
1. Song, Junha, et al. "Ecotta: Memory-efficient continual test-time adaptation via self-distilled regularization." Proceedings of the IEEE/CVF Conference on Computer Vision and Pattern Recognition. 2023.
2. Lee, Daeun, Jaehong Yoon, and Sung Ju Hwang. "BECoTTA: Input-dependent Online Blending of Experts for Continual Test-time Adaptation." International Conference on Machine Learning. PMLR, 2024.

**Experimental Designs Or Analyses:**

Refer to the weaknesses and questions.

**Methods And Evaluation Criteria:**

The proposed approach shows improved performance for the compared ViT-B/16 architecture.
However, the lack of experiments on CNNs, such as WideResNet-28,  ResNeXt-29 and ResNet-50 that have been widely used by the state-of-the-art continual TTA methods, hurts the applicability and utility of the proposed approach.
Moreover, several state-of-the-art approaches, such as EcoTTA [1], EATA [2], BeCoTTA [3], and PETAL [4], are missing from experimental comparisons.

References:
1. Song, Junha, et al. "Ecotta: Memory-efficient continual test-time adaptation via self-distilled regularization." Proceedings of the IEEE/CVF Conference on Computer Vision and Pattern Recognition. 2023.
2. Niu, Shuaicheng, et al. "Efficient test-time model adaptation without forgetting." International conference on machine learning. PMLR, 2022.
3. Lee, Daeun, Jaehong Yoon, and Sung Ju Hwang. "BECoTTA: Input-dependent Online Blending of Experts for Continual Test-time Adaptation." International Conference on Machine Learning. PMLR, 2024.
4. Brahma, Dhanajit, and Piyush Rai. "A probabilistic framework for lifelong test-time adaptation." Proceedings of the IEEE/CVF Conference on Computer Vision and Pattern Recognition. 2023.

**Other Comments Or Suggestions:**

1. Line 325: Section 4.2 is cited inside the same section.
2. Include references to hyperparameter details and tuning in the main paper.
3. This sentence is not grammatically correct: "The principal idea is to explicitly enhance the prediction complexity of a sample by masking objects that domain invariant features."

**Other Strengths And Weaknesses:**

**Strengths**
* An interesting idea to improve data augmentation via incremental masking for Continual TTA
* The experiments show improved performance on ViT

**Weaknesses**
* An elaborate detail about explicit mask chaining mechanisms seems to be lacking. This is a key element of the proposed approach.
* Lack of experiments on CNNs, such as WideResNet-28, ResNeXt-29, and ResNet-50, that have been widely used by the state-of-the-art continual TTA methods, hurts the applicability/utility of the proposed approach.
* Details about hyperparameters such as "M and N involved in mask ratios and mask chains" are missing
* Exploiting the self-attention structure limits the applicability to ViT architectures.

**Questions For Authors:**

1. What are the values of M and N involved in mask ratios and mask chains? Are these values tuned?
2. How does incrementally masking content involve only the content containing the domain-invariant information? How is the self-attention structure utilized for doing this?
3. Do they authors have any experiments on CNNs, such as WideResNet-28,  ResNeXt-29 and ResNet-50, that have been widely used by the state-of-the-art continual TTA methods?
4. How is the hyperparameter λ in Equation 5 tuned? In Appendix E, is the error computed on the test split itself, on which the performance is reported?
5. It is mentioned "we set λ = 1 and m = 0", and the Figure 12 shows that the margin does not matter. Why is it so, do the authors have any explanation or intuition about it?
6. Is explicit mask chaining idea adopted from other existing literature? If so, a citation with better description will improve clarity.

**Relation To Broader Scientific Literature:**

The key contribution of this paper is utilizing well established ranking loss using an incremental masking along with mask consistency loss which seems to improve CTTA.

**Theoretical Claims:**

NA

---

> ### Author Rebuttal · Authors · 2025-03-30
>
> We appreciate your positive reviews and valuable suggestions. We address your concerns and questions below in detail.
> >**1. Values of M and N involved in mask ratios and chains**
>
> A1. We provide the ablation test regarding hyperparameters of the mask. Although the best accuracy was achieved when N=3, we used the combination of M_N ={0,5%,10%} to strike a balance between sensitivity to the masking ratio, increased computational complexity, and accuracy.
> M_N (N=1)|{0,5%}|{0,10%}|{0,15%}
> ---|---|---|---
> Error|40.6|39.7|39.5
> M_N (N=2)|{0,5%,10%}|{0,10%,20%}|{0,15%,30%}
> Error|39.4|39.2|39.4
> M_N (N=3)|{0,5%,10%,15%}|{0,10%,20%,30%}|{0,15%,30%,45%}
> Error|38.9|39.4|40.0
>
> >**2. Details of incremental masking and its references**
>
> A2. Our research is inspired by [R2A], which demonstrates that the self-attention mechanism in ViTs naturally clusters tokens with similar contexts, merging semantically similar tokens. Building on this, we follow the implementation of [R2B], which improves ViT efficiency by omitting background tokens. Specifically, for TTA, we compute attention scores by averaging the similarity between the class token's query and the image tokens' keys across all heads in the final multi-head self-attention layer. Tokens with high attention scores are identified, and forward passes for the corresponding foreground tokens are selectively omitted. Based on the findings of [R2A] and [R2B], we confirm that efficient masking is achievable through the self-attention structure. By progressively masking foreground regions, we control the prediction difficulty and provide a structured approach to address the challenges of continual TTA.
>
> [R2A] Token Merging: Your ViT But Faster, ICLR2023.
> [R2B] The Role of Masking for Efficient Supervised Knowledge Distillation of Vision Transformers, ECCV2024.
>
> >**3. Experiments on CNNs**
>
> A3. Thank you for suggesting experiments on CNNs. While our method leverages the powerful self-attention mechanism in ViTs, it can be easily extended to other networks as long as the difficulty can be structured through explicit mask chaining. To explore this, we provide two additional experiments: one that uses the activation of the final CNN feature map (FA), and another that adjusts masked pixels using Grad-CAM. For CIFAR10-C, we present results based on EcoTTA using WRN-28, and for CIFAR100-C, we report results from BECoTTA using WRN-40.
> Dataset|EATA|EcoTTA|BECoTTA|Ours(FA)|Ours(Grad-CAM)
> ---|---|---|---|:---:|:---:
> CIFAR10C|18.6|16.8|-|16.9|**16.5**
> CIFAR100C|37.1|36.4|35.5|**34.5**|34.6
>
> >**4. Hyperparameter $\lambda$**
>
> A4. We also share the concern regarding tuning hyperparameters on the test set in TTA. In fact, there exist combinations of lambda and margin that yield higher performance, but we use 1 and 0 as default values for lambda and margin, respectively. Additionally, to verify the robustness of our method to hyperparameters, Fig.12 highlights that our method maintains consistent performance across a wide range of values rather than being sensitive to specific settings.
>
> >**5. Margin $m$**
>
> A5. The margin is related to whether entropy minimization is applied to each sample. A higher margin $m$ leads to entropy minimization for more samples, enabling faster adaptation, while a lower $m$ prioritizes stability. This corresponds to the stability-plasticity trade-off commonly observed in continual learning. In line with the goal of our proposed method, we adopt a margin value of 0 to mitigate overconfidence.
> m|0|0.1|0.2|0.3
> ---|---|---|---|---
> Error|39.2|38.8|38.8|38.9
> ECE|8.7|10.2|10.8|11.0
>
> >**Weakness**
>
> - [Mask chaining mech.] Please refer to A2.
> - [CNN experiments] Please refer to A3.
> - [Hyperparameter M and N] Please refer to A1.
> - [Self-attention structure limitation for ViT] While our method leverages the self-attention mechanism, it is not limited to standard ViT architectures. As demonstrated in Response 8 to Reviewer f4Hw, our approach is applicable to various transformer backbones such as MobileViT and SwinTransformer. Furthermore, we also provide CNN-based experiments in Response A3.
>
> >**Minor**
>
> - Thanks for your suggestions on typos and references. We will revise it for better readability.

---

> > ### Comment · Reviewer_MEbz · 2025-04-02
> >
> > Thanks for responding to the queries.
> >
> > I would like to point out that CoTTA [1] reports an error rate of 16.2 on the CIFAR10-to-CIFAR10C dataset using the WRN-28 backbone (3. Experiments on CNNs), so it is doing better than the proposed approach in this experiment.
> >
> > As pointed out, tuning hyperparameters on the test set is a matter of concern.
> >
> > Other than this, I do not have any questions at this moment.
> >
> > Thanks.
> >
> > References:
> > 1. Wang, Qin, et al. "Continual test-time domain adaptation." Proceedings of the IEEE/CVF Conference on Computer Vision and Pattern Recognition. 2022.

---

> > > ### Author Response · Authors · 2025-04-04
> > >
> > > We sincerely thank the reviewer MEbz for your time and valuable feedback on our work.
> > >
> > > >**A1. CNNs experiments**
> > >
> > > Thank you for your in-depth suggestions. We will add the results for CoTTA to the final version of the paper. We would like to emphasize that our method has been extensively evaluated on ViT to confirm that it achieves successful results, and that our work on utilizing transformer structures has broader applicability, including its use in multimodal systems such as CLIP. Our message regarding CNN experiments is that our method can be extended to CNNs if we establish a ranked structure with explicit difficulty control, which is the basic philosophy of our mechanism. In this respect, we verified the scalability of our method by following the experimental protocols of EcoTTA and BECoTTA.
> > >
> > > >**A2. Concern about hyperparameters**
> > >
> > > We fully understand and appreciate your concerns regarding the hyperparameters. To address this, we confirmed that our method is robust across a wide range of hyperparameters, including $\lambda$, $m$, and $M$_$N$, through extensive investigations in Fig.12 and our previous response (A1). We adopted the basic values by setting the coefficient for the loss function to 1 and the margin for the ranking loss to 0, using the same settings consistently across ImageNetC, CIFAR10C, and CIFAR100C.
> > > We would like to emphasize that we did not tune the hyperparameters specifically for the test data to enhance performance. Furthermore, as shown in Figure 6, we investigated the learning rate across a broad range and observed stable adaptation of our method. This supports our claim that the proposed method adapts reliably across a wide range of hyperparameter settings.
> > >
> > > We again thank you for your valuable and constructive feedback.

---

### Official Review · Reviewer_f4Hw · 2025-03-14

**Overall Recommendation:** 2

**Summary:**

This work introduces two novel loss functions for CTTA, which utilize different views of the samples to enforce consistency alignment while preserving their relative ranking. Experimental results demonstrate the effectiveness of the proposed approach.

**Claims And Evidence:**

Yes

**Essential References Not Discussed:**

The work would benefit from further exploration and discussion of key references.

**Experimental Designs Or Analyses:**

Missing comparisons with strong baseline for CTTA:
1. EATA 2. Roid 3. Test-Time Ensemble via Linear Mode Connectivity

Lack of important ablation and discussion about the key hyperparameters, N and M_N in Eqn. (3).

Inefficiency in computation compared to the entropy-based method, and potentially much higher memory consumption by using more bp per samples.

Lack of justification of resizing images to 384x384 on CIFAR, which significantly increases the computation burden but seems only beneficial to the proposed method that requires masking.

Limited evaluations on only ViT-Base. Suggest to add more evaluations on architectures like MobileViT, SwinTransformer.

Random masking without a projector can be unstable, but the paper's results do not show the variance in performance.

**Methods And Evaluation Criteria:**

Lack of novelty. the idea of consistency learning between strong and weak augmentation has been thoroughly explored by existing TTA works like [1,2]. Meanwhile, these methods are not discussed or compared.
1. Contrastive Test-Time Adaptation
2. Revisiting Realistic Test-Time Training: Sequential Inference and Adaptation by Anchored Clustering Regularized Self-Training


Lack of important details in method details like foreground masking in figure 10, where does the paper introduce foreground masking?

The propose method may lead to biased attention towards the background content, compared to source, as evidenced by the 3rd row of Figure 8.

**Other Comments Or Suggestions:**

I would raise the score if the authors address all concerns

**Other Strengths And Weaknesses:**

The paper claims to address overconfidence but provides no numerical experiment on metrics like false positive rate or ECE.

**Questions For Authors:**

Questions: Is this fully TTA? More comparisons on ViT weights that is not pre-trained with MAE, like Moco-V3, to support this.

Overall, the technical contribution is limited and some key technical details are unclear with insufficient evaluations and ablations.

**Relation To Broader Scientific Literature:**

NA

**Theoretical Claims:**

NA

---

> ### Author Rebuttal · Authors · 2025-03-29
>
> We appreciate your constructive comments. We address all the concerns and provide new experiments to support our contributions.
> >**1. Novelty**
>
> Compared with [1,2], our novelty is summarized;
> - [1] proposed a contrastive learning method based on strong-weak augmentations, but our method addresses the problem within a ranked structure through progressive augmentation, and it differs in that ours does not require an additional momentum encoder and does not rely on random augmentation.
> - [2] analyzed that using the prediction from weak augmentation as a pseudo label to improve the prediction of strong augmentation is not effective. It aligns with the argument of CoTTA that data augmentation without considering sample characteristics is not suitable for handling abrupt distribution shifts. Ours applies masking on objects as a domain-invariant property and maintains a ranked structure, enabling stable adaptation by reducing discrepancies in prediction distributions.
>
> Therefore, ours has its own novelty compared with [1,2].
>
> >**2. Foreground masking**
>
> The detailed descriptions on foreground masking are found at Sec 3.2. Briefly, for the final self-attention layer of the transformer, attention scores are computed based on the similarity between the class token's query and the image tokens' keys. Using these scores, we can efficiently and naturally mask the pixels where objects are located without any extra modules.
>
> >**3. Biased attention**
>
> Due to differences in object sizes across samples, a bias toward the background can occur when small objects occupy a relatively small portion of the pixels. Based on our observations in Fig. 3, to alleviate this concern, we apply low masking ratios of 10% and 20% where a linear relationship between entropy and error is maintained. Note that our goal is to find a simple yet efficient solution without additional computational cost.
>
> >**4. Additional comparison**
>
> We provide the following table as requested. Note that TTE [3] and its code are not presented yet.
> Method|EATA|SAR|PETAL|Roid|Ours
> ---|---|---|---|---|---
> ImageNetC|41.3|45.2|52.3|41.4|**39.2**
>
> [3] Test-Time Ensemble via Linear Mode Connectivity, ICLR2025.
>
> >**5. Ablation on mask**
>
> We provide the ablation study regarding hyperparameters of the mask. Although the best accuracy was achieved when N=3, we used the combination of M_N ={0,5%,10%} to strike a balance between sensitivity to the masking ratio, increased computational complexity, and accuracy.
> M_N (N=1)|{0,5%}|{0,10%}|{0,15%}
> ---|---|---|---
> Error|40.6|39.7|39.5
> M_N (N=2)|{0,5%,10%}|{0,10%,20%}|{0,15%,30%}
> Error|39.4|39.2|39.4
> M_N (N=3)|{0,5%,10%,15%}|{0,10%,20%,30%}|{0,15%,30%,45%}
> Error|38.9|39.4|40.0
>
> >**6. Comparison with entropy-based method**
>
> Similar to entropy-based methods, we use a single backpropagation per sample. Recent TTA approaches, including EATA, update parameters only using low-entropy samples, thereby improving learning efficiency. Furthermore, parameter-free methods such as T3A and LAME adapt to new domains by aligning logits using optimization algorithms or historical memory without backpropagation. Each of these methods is being studied independently and is based on a different underlying philosophy. We acknowledge that our method is slightly less computationally efficient compared to these approaches. Nevertheless, we believe that utilizing uncertain samples in alignment with the motto of continual adaptation is necessary for broader applicability in diverse environments. We think that these three works will converge by complementing each other’s strengths and weaknesses, and we believe our contribution can narrow the gap between them.
>
> >**7. Resolution**
>
> For a fair comparison of SOTA, we followed the experimental settings of ViDA and Continual-MAE, using 384×384 for CIFAR, and ours was built on Continual-MAE. We agree that higher resolution can offer advantages for masking, and to clarify this point, we provide a 224x224 experiment.
> CIFAR10C|Source|Tent|CoTTA|Continual-MAE|Ours
> ---|:---:|:---:|:---:|:---:|:---:
> ViT-B/16-224|40.1|36.0|39.2|16.8|**7.9**
>
> >**8. Transformer backbone**
>
> We provide experiments using different transformer backbones.
> ImageNetC|Source|Tent|CoTTA|ViDA|Ours
> ---|---|---|---|---|---
> Mobile-ViT-S|75.28|75.61|75.72|75.27|**74.28**
> SwinTrans.-B|59.26|73.17|46.84|57.84|**46.56**
>
> >**9. Mask**
>
> Though our method employs a masking strategy, it is used solely to control the difficulty of prediction, and thus does not require a projector during the masking process or a decoder to reconstruct the masked regions, and it is relatively free from instability.
>
> >**10. Overconfidence**
>
> We provide a table for calibration error analysis.
> ImageNetC|Tent|ViDA|Ours
> ---|---|---|---
> ECE (low)|12.6|14.6|**8.7**
>
> >**11. MoCo-v3**
>
> Thanks for your suggestion to utilize MoCo-V3, not trained with labels on source domain.
> ImageNetC|Source|Tent|CoTTA|SAR|PETAL|Ours
> ---|---|---|---|---|---|---
> Moco-v3|76.5|76.6|78.2|75.4|78.2|**65.7**

---

> > ### Comment · Reviewer_f4Hw · 2025-04-03
> >
> > Thank you for the detailed response from the authors. I have carefully read all responses. However, the ablations on hyperparameter N and M_N fail to convey the effectiveness of the proposed ranked structure, where the method achieves 39.5% with N=1 and 39.2% with N=2. Further, how is the overconfidence compared to Source and SAR?
> >
> > Moreover, I have further questions for the foreground masking approach. 1) While foreground masking can create difficult mask using a low masking ratio, it ignores the potential benefit of learning background-invariant features. To create difficult masks, I assume using the random masking strategy, while increasing the mask ratio to 40% (or above) would work fine. I suggest tuning the mask ratios of different masking strategy in Figure 10 for a fair comparison. 2) How to create foreground masking for transformers that do not use classification token, e.g., averaging the features of all tokens as the input of the classifier.

---

> > > ### Author Response · Authors · 2025-04-04
> > >
> > > We thank the reviewer f4Hw for your thoughtful and constructive feedback.
> > >
> > > >**A1. Ablation on M_N**
> > >
> > > In the course of our study, we first confirmed that our ranking-based solution is working when applied with a single mask (e.g., N=1). Notably, our error rate of 39.5% (e.g., M_N={0,15%}) represents a 3% improvement over the recent state-of-the-art result of 42.5% by Continual-MAE, supporting the effectiveness of our ranked structure.
> > >
> > > We furthermore investigated whether our approach could be generalized to a chained masking structure for N = 2, 3, …. We achieved the error rates of 39.2% for N = 2 and 38.9% for N = 3 (please, remember that Continual-MAE is still 42.5%), which validates our method that generalizes across various values of N.
> > >
> > > Finally, we selected M_N by considering the trade-off between performance and computational cost as N increases. The proposed method not only demonstrates its effectiveness at N = 2, but also offers flexibility in selecting appropriate N values depending on computational budget and performance requirements.
> > >
> > > >**A2. Additional ECE comparison**
> > >
> > > Additionally, we include a comparison table of the ECE between the source, SAR, and our method as you requested. We observe a trend in which the ECE tends to increase with performance improvement over the initial source model. Notably, our method maintains a low ECE while achieving low error rates.
> > >
> > > ImageNetC|*Source*|Tent|*SAR*|ViDA|Ours
> > > ---|:---:|:---:|:---:|:---:|:---:
> > > ECE (%) |5.3|12.6|10.3|14.6|8.7
> > > Error (%) |55.8|51.0|45.2|43.4|39.2
> > >
> > > >**A3. Masking strategy**
> > >
> > > We acknowledge that the foreground masking strategy using self-attention is not the only possible approach. It is one of several design choices, and **our primary contribution lies in utilizing the ranked structure induced by an explicit augmentation.**
> > >
> > > In particular, our usage of foreground masking was motivated by two main considerations:
> > > (i) it can be efficiently applied without any additional complex modules, as it directly utilizes the self-attention mechanism; and
> > > (ii) it ensures the construction of ranking relationships through explicitly designing the prediction difficulties.
> > >
> > > - 1 ) Our concern with random masking was that when a large proportion of masking tokens are located in the background, it may inadvertently focus attention on the foreground, potentially failing to preserve the intended ranking relationships. This potential risk is evident in the experiment on random masking in CIFAR100C shown in Figure 10, where effective adaptation is initially achieved but a dramatic performance drop occurs from the 5th task, e.g., glass blur.
> > >
> > > - 2 ) As you suggested, utilizing feature attention (FA) is a promising approach to extend our method to architectures beyond self-attention structures. This is confirmed to be a feasible solution, as demonstrated in Reviewer MEbz's A3, which is an extension of our method to CNNs.
> > >
> > > For clarity, we present the results using 40-60% Random Masking (RM) and Feature Attention (FA) in the table below. As a detailed implementation of FA, the attention map is computed as the average of the L2 norm of the features over all image tokens.
> > >
> > > Method|Ours-RM (40%)|Ours-RM (50%)|Ours-RM (60%)|Ours-FA|Ours
> > > ---|:---:|:---:|:---:|:---:|:---:
> > > Error (%) |40.5|41.1|42.1|39.9|39.2
> > >
> > > Finally, we would like to clarify a typo in our previous response (A5) regarding the ablation on masking. M_N = {0, 10%, 20%} was used in the experiments. We hope that our response has helped address your concerns.
> > >
> > > Once again, we sincerely appreciate your in-depth review of our paper and your insightful suggestions for improving the quality of our paper.

---

### Decision · Program_Chairs · 2025-05-01

**Decision:**

Accept (poster)

**Comment:**

Five experts in the field reviewed this paper, and their recommendations are two Weak Rejects, one Weak Accept, and two Accepts after the rebuttal phase. Overall, the reviewers appreciated the paper because it provides a well-motivated solution to a challenge of model collapse under entropy minimization in continual TTA. Entropy ranking via progressive masking is intuitive and empirically validated. The proposed masked consistency and entropy ranking loss stabilize adaptation during deployment without requiring access to source data or complex regularization.

The reviewers raised some concerns regarding the following weaknesses:
- The proposed REM approach (e.g., strong/weak augmentation) has incremental contributions in TTA, although the structured ranking of entropy is an original component.
- REM requires multiple forward passes per sample, leading to higher computational cost versus TTA methods like TENT. Computational efficiency and scalability remain a concern for resource-constrained or real-time deployments.
- Improvements over standard augmentation strategies (e.g., random cropping, affine transformations) is unclear. How much of the gain is due to the specific design versus broader augmentation and entropy combinations.
The authors are encouraged to consider these comments when revising the paper.

I recommend accepting this paper based on the reviewers’ feedback and the authors’ detailed rebuttal. The rebuttal addresses most major concerns with extended experiments, analysis, and discussions of REM limitations. While the authors provided some intuitions, the work does not include strong theoretical justification to support the mitigation of model collapse. While some theoretical depth is still lacking, the method’s effectiveness and relevance to an active research area justify its inclusion. The paper can make valuable contribution to the TTA literature with minor revision. The proposed method is highly integrated with ViT models using self-attention-based foreground masking. The authors extended the approach to CNNs via Grad-CAM and attention-based feature maps. The empirical validation is convincing based on ImageNet-C, CIFAR10-C, CIFAR100-C, and ImageNet-3DCC in continual TTA under both TTA settings. The authors’ extended results include multiple ViT backbones and some CNN-based variants. Ablation studies show robustness to hyper-parameter values. Additionally, REM improves both classification accuracy and calibration, showing that it may be suitable for real-world deployment.